



# Albedo susceptibility of Northeastern Pacific stratocumulus: the role of covarying meteorological conditions

Jianhao Zhang[1,2,3], Xiaoli Zhou[1,3], and Graham Feingold[1]

[1]Chemical Sciences Laboratory, National Oceanic and Atmospheric Administration (NOAA), Boulder, CO, USA
[2]National Research Council, National Academies of Sciences, Engineering, Medicine (NASEM), Washington DC, USA
[3]Cooperative Institute for Research in Environmental Sciences (CIRES), University of Colorado, Boulder, CO, USA

**Correspondence:** Jianhao Zhang (jianhao.zhang@noaa.gov)

**Abstract.** Quantification of the radiative adjustment of marine low-clouds to aerosol perturbations, regionally and globally, remains the largest source of uncertainty in assessing current and future climate. An important step towards quantifying the role of aerosol in modifying cloud radiative properties is to quantify the susceptibility of cloud albedo and liquid water path (LWP) to perturbations in cloud droplet number concentration ($N_d$). We use 10 years of space-borne observations from the polar-orbiting Aqua satellite, to quantify the albedo susceptibility of marine low-clouds over the northeast (NE) Pacific stratocumulus region to $N_d$ perturbations. Overall, we find a low-cloud brightening potential of $20.8 \pm 0.96$ W m$^{-2}$ ln($N_d$)$^{-1}$, despite an overall negative LWP adjustment for non-precipitating marine stratocumulus, owing to the high occurrence (37% of the time) of thin non-precipitating clouds (LWP $< 55$ g m$^{-2}$) that exhibit brightening. In addition, we identify two more susceptibility regimes, the entrainment-darkening regime (36% of the time), corresponding to negative LWP adjustment, and the precipitating-brightening regime (22% of the time), corresponding to precipitation suppression. The influence of large-scale meteorological conditions, obtained from the ERA5 reanalysis, on the albedo susceptibility is also examined. Over the NE Pacific, clear seasonal covariabilities among meteorological factors related to the large-scale circulation are found to play an important role in grouping favorable conditions for each susceptibility regime. Our results indicate that, for the NE Pacific stratocumulus deck, the strongest positively susceptible cloud states occur most frequently for low cloud top height (CTH), the highest lower-tropospheric stability (LTS), low sea-surface temperature (SST), and the lowest free-tropospheric relative humidity (RH$_{ft}$) conditions, whereas cloud states that exhibit negative LWP adjustment occur most frequently under high CTH and intermediate LTS, SST, and RH$_{ft}$ conditions. The warm rain suppression driven cloud brightening is found to preferably occur either under unstable atmospheric conditions (low LTS) or high RH$_{ft}$ conditions that co-occur with warm SST. Mutual information analyses reveal a dominating control of LWP, $N_d$ and CTH (cloud state indicators) on low-cloud albedo susceptibility, rather than of the meteorological factors that drive these cloud states.

## 1 Introduction

Changes in aerosol concentrations in the marine boundary layer, of either natural or anthropogenic origin, can lead to significant changes in the brightness of marine low-level clouds. Examples of aerosol induced changes in cloud reflectivity are observed in aerosol perturbations associated with natural causes, such as volcanic eruptions (e.g. Gassó, 2008; Yuan et al., 2011; Malavelle





et al., 2017), and anthropogenic sources across the globe, such as ship emissions, wildfires, and power plants (Toll et al., 2019). Among anthropogenic sources, shiptracks – bright linear cloud features associated with particle emissions (Coakley et al., 1987) – have been used to improve our understanding of cloud responses to aerosol perturbations. The routine and frequent occurrence of global shipping traffic, and constant meteorological conditions in- and out-of-shiptrack make them a 'natural laboratory' to improve our understanding of cloud responses to aerosol perturbations. Studies based on satellite observations
(e.g. Coakley and Walsh, 2002; Gryspeerdt et al., 2019b; Chen et al., 2012; Christensen and Stephens, 2011; Christensen et al., 2014) and idealized frameworks such as large-eddy simulations (e.g. Wang et al., 2011; Hill et al., 2009), have been used to quantify/constrain the global aerosol radiative effect (e.g. Diamond et al., 2020). However, to date, our ability to narrow down estimates of climate sensitivity is still limited by uncertainties related to quantifying the radiative adjustment of marine low-clouds to the anthropogenic aerosol (Boucher et al., 2013).

For non-precipitating warm clouds exhibiting constant liquid water path (LWP), increases in aerosol concentration result in increases in droplet concentration leading to smaller droplets that make the cloud more reflective (the Twomey effect; Twomey, 1974, 1977). These processes occur at short timescales (order 5 – 10 min). However, LWP is not always constant: LWP adjustments were first suggested to exist in precipitating marine warm clouds: an increase in $N_d$ leads to smaller cloud droplets that are less likely to grow by collision-coalescence to precipitation-sized raindrops under the same environmental
conditions (Albrecht, 1989). The result is a reduction in the loss of cloud water due to precipitation, which then leads to an increase in LWP, that enhances cloud brightening associated with the smaller drops.

More recently, negative LWP adjustments in non-precipitating stratocumulus have also been identified: (i) the reduced droplet sizes decrease the sedimentation flux at stratiform cloud-tops, enhancing the evaporative and radiative cooling and thereby the entrainment rate at cloud tops (the sedimentation-entrainment feedback; Ackerman et al., 2004; Bretherton et al.,
2007); (ii), smaller cloud droplets evaporate faster, leading to stronger cooling and more turbulent mixing at cloud-tops, which then causes more evaporation, creating a positive feedback loop, known as the evaporation-entrainment feedback (Wang et al., 2003; Xue and Feingold, 2006; Jiang et al., 2006). Both these entrainment-feedbacks reduce cloud LWP in response to the increased concentration of smaller droplets, resulting in less reflective clouds and hence a warming relative to a cloud with constant LWP. A strong offsetting warming effect from the negative LWP adjustment is evident in both observational studies
(e.g. Possner et al., 2020; Gryspeerdt et al., 2019a, 2021) as well as large eddy simulation (e.g. Wang et al., 2003; Ackerman et al., 2004; Xue et al., 2008). The timescale associated with these negative LWP adjustments is $t \approx 20$ h (Glassmeier et al., 2021). Because shiptracks exist for only 7 h, and are likely to be sampled on average after 3.5 h, a generalization of shiptrack characterized aerosol-cloud interactions to estimates of anthropogenic aerosol climate forcing may be substantially overestimated because the shiptrack has not existed for long enough to manifest full negative LWP adjustment.

Moreover, despite routine shipping traffic, ship tracks are only rarely observed over major shipping corridors (only 0.002% of the total ocean-going ship traffic; Campmany et al., 2009), in part due to the narrow range of meteorological conditions required for these bright tracks to form (Durkee et al., 2000). This suggests that the coupled large-scale meteorology and the associated cloud states have a strong impact on the susceptibility of low-clouds to aerosol perturbations. Several studies have tried to constrain the uncertainties in LWP and reflectance adjustments based on cloud states and large-scale meteorological



conditions using satellite observations (e.g. Chen et al., 2014; Douglas and L'Ecuyer, 2019; Possner et al., 2020) and found strong meteorological controls on cloud state and cloud albedo susceptibility to aerosol perturbations across the globe: regions with relative dry and unstable conditions tend to be characterized by cloud darkening in response to increased aerosol loading, whereas clouds in stable and moist regions tend to brighten in response to increased aerosol concentrations.

Here, we quantify relationships between cloud albedo ($A_c$) and $N_d$ using satellite-retrieved cloud properties and radiative
fluxes over the northeast Pacific marine stratocumulus deck, one of the regions contributing most strongly to the overall cooling of the Earth by reflecting incoming solar radiation (Klein and Hartmann, 1993). One should note that this approach of analyzing a composite of satellite snapshots of cloud fields ought to be restricted to inferring relationships between cloud properties from a climatological perspective where a sufficient amount of sampling of a time-space varying system creates a robust characterization of the relationships between quantities that describe the system. This contrasts with approaches aimed at the evolution
of cloud systems, i.e. quantifying the time derivatives of cloud properties, through tracking properties of the system, either by numerical simulations or temporally-resolved satellite observations that take consecutive snapshots of an evolving cloud field (e.g. Glassmeier et al., 2021; Gryspeerdt et al., 2014). Cloud albedo susceptibilities to $N_d$ perturbations are approximated by regressed linear relationships between $N_d$ and $A_c$ of a given satellite snapshot, similar to Painemal (2018), but under covarying meteorological conditions, by applying constrained cloud states and meteorological conditions obtained from the ERA5 reanal-
yses. Furthermore, we examine and characterize meteorological conditions that favor the occurrence of susceptible and less susceptible conditions, i.e. the potential for a warming or a cooling effect. Datasets and the methodology used in this study are described in Section 2 and Section 3, respectively. The Result Section (Section 4) presents a characterization of climatological relationships between $A_c$, LWP and $N_d$, an examination of cloud albedo susceptibility and susceptibility regimes in a LWP-$N_d$ space, and the role of meteorological conditions in cloud albedo susceptibility. Key findings and conclusions are summarized
in Section 5.

## 2   Datasets

This study focuses on an area of 10° by 10° (120–130°W, 20–30°N) over the subtropical Northeast (NE) Pacific stratocumulus region, corresponding to an area of regional maximum in annual stratus cloud amount, which is the same region examined in Klein and Hartmann (1993). Marine low-cloud properties and shortwave (SW) radiative measurements are retrieved from
the MODerate resolution Imaging Spectroradiometer (MODIS) (Platnick et al., 2003) and the Clouds and the Earth's Radiant Energy Systems (CERES; Wielicki et al., 1996) sensors onboard the Aqua satellite (overpass ∼1:30 pm local time), obtained from the CERES Single Scanner Footprint (SSF) product Edition 4 (level 2; Su et al., 2015). Top-of-atmosphere (TOA) SW fluxes, including incoming solar radiation ($SW_{TOA_{dn}}$) and reflected SW flux ($SW_{TOA_{up}}$), are derived from the Single Scanner at a CERES footprint resolution of 20 km (Loeb et al., 2005; Su et al., 2015), which are then used to calculate cloud SW albedo
($A_c$) as follows:

$$A_c = \frac{(A_{all} - A_{clr}(1 - f_c))}{f_c} \qquad (1)$$





where $A_{all}$ is scene albedo (all-sky albedo), defined as the ratio of $SW_{TOA_{up}}$ to $SW_{TOA_{dn}}$, $A_{clr}$ is the SZA dependent ocean albedo (clear-sky albedo), derived from the scene albedo under clear sky conditions over the study area, and $f_c$ is the cloud fraction.

MODIS cloud properties, including cloud optical depth ($\tau$), cloud top effective radius ($r_e$), $f_c$, LWP, cloud effective temperature, and cloud top height (CTH) are retrieved using the CERES-MODIS algorithm at MODIS pixels and then aggregated to CERES footprint resolution and scanning pattern (Minnis et al., 2011b, a). Retrieval of $r_e$ is based on the 3.7-$\mu m$ channel, which has been shown to be less affected by retrieval biases than the 2.1-$\mu m$ and 1.6-$\mu m$ channels (Grosvenor et al., 2018). $N_d$ is calculated following Grosvenor et al. (2018) as

$$N_d = \frac{\sqrt{5}}{2\pi k} \left( \frac{f_{ad} c_w(T,P)\tau}{Q_{ext}\rho_w r_e^5} \right)^{1/2} \qquad (2)$$

where $k$ is a parameter representing the width of the modified gamma droplet distribution (assumed to be 0.8), $f_{ad}$ is the adiabatic fraction (assumed to be 0.8), $c_w$ is the condensation rate, which is a function of temperature (T) and pressure (P) (Grosvenor and Wood, 2014), calculated using CERES-MODIS cloud effective temperature at a constant pressure of 900 hPa, $Q_{ext}$ is the extinction efficiency factor, approximated by its asymptotic value of 2 (Grosvenor et al., 2018), and $\rho_w$ is the density

of liquid water. In addition, $N_d$ is only calculated for CERES footprints with $f_c > 0.99$ (overcast footprints), cloud effective temperature greater than 273 K (to exclude mixed-phase and ice clouds), CTH less than 3 km, $\tau > 3$, and $r_e > 3 \ \mu m$, and solar zenith angle (SZA) < 65°, to minimize retrieval biases (Painemal et al., 2013; Grosvenor and Wood, 2014; Grosvenor et al., 2018). Furthermore, footprints with a calculated $N_d$ greater than 600 cm$^{-3}$ are discarded to avoid highly unrealistic $N_d$ retrievals.

Meteorological conditions, including sea surface temperature (SST), sea level pressure (SLP), vertical velocity at 700 hPa ($\omega 700$), and temperature, humidity, and wind profiles, are obtained from the European Centre for Medium-Range Weather Forecasts (ECMWF) fifth-generation atmospheric reanalysis (ERA5; Hersbach et al., 2020), available every hour at 0.25° spatial resolution. Lower-tropospheric-stability (LTS) is calculated as the difference in potential temperature between 700 hPa and 1000 hPa. Free-tropospheric relative humidity ($RH_{ft}$) is defined as the the mean relative humidity between inversion top

and 700 hPa, following Eastman and Wood (2018).

## 3    Methods

The direct causal relationship between aerosol and cloud properties, including the brightness of a cloud field, is often obscured by the confounding local meteorological conditions that have influences on both the aerosol and cloud properties (e.g. Mauger and Norris, 2007; Gryspeerdt et al., 2014). Gryspeerdt et al. (2016, 2019a) have shown that a good mediating variable, namely

$N_d$, can help reveal the casual relationship between $N_d$–$f_c$ and $N_d$–LWP, by eliminating the influence of local meteorology on the causal pathway, using conditional probabilities derived from satellite observations. This is rooted in the so called Calculus of Actions, introduced by Pearl (1994), who showed that an observed relationship (seeing) can be used to determine the





outcome of an action (doing or causality) when no confounding effects are present, meaning the causal parents of X, $pa_i$s, are independent of the outcome of an action, Y, given a causal network G(X, Y, $pa_i$).

This work adopts the same logical approach, that is to infer casual relationships from observed $N_d$–$A_c$ relationships derived from satellite snapshots of cloudy scenes under conditions where the influence of confounding factors on the casual pathway are minimized at the scale at which the observational relationships/associations are deduced. We achieve this by deriving $N_d$–$A_c$ relationships within a limited space-time frame, that is $1° \times 1°$ area at 1:30 local afternoon, such that the confounding large-scale meteorology is assumed constant within the selected space-time frame and thereby independent of the sub-degree (20 km

footprint) varying cloud properties from which we derive the relationships. Moreover, although joint histograms built upon a composite of satellite snapshots better determine the conditional probability distributions describing a non-linear relationship, for instance the $N_d$–LWP relationship (Gryspeerdt et al., 2019a), if we narrow our lens spatially and temporally down to a single satellite snapshot at a given time over a $1° \times 1°$ area, slopes of linear regressions can be deterministic/representative of the local-transient relationships between two cloud properties, similar to the finite difference method used to approximate the

local derivatives. Hence, in this work, we use slopes derived from least squares linear regressions of $N_d$–$A_c$ relationships in ln-ln space, sampled by the MODIS and CERES sensors onboard the polar-orbiting Aqua satellite, on a $1°$ grid, to infer cloud albedo susceptibility ($S_0$), a casual relationship, represented as follows:

$$S_0 = \frac{dln(A_c)}{dln(N_d)}.$$  (3)

The conditions for such an approach to be carried out are met when at least 5 $N_d$ retrievals are available with only a single-

layer liquid cloud being present, and the absolute value of the correlation coefficient between $A_c$ and $N_d$ is greater than 0.2, within the $1°$ grid. The correlation coefficient requirement helps us remove cloudy scenes where regressed slopes are highly questionable and thereby unreliable.

    Furthermore, the cloud albedo sensitivity to $N_d$ perturbations is converted to a radiative sensitivity as an intermediate step towards quantifying the radiative forcing, by multiplying the albedo susceptibility by gridbox low-cloud fraction and the

incoming solar flux. This is termed radiative susceptibility ($F_0$) hereafter, equivalent to a radiative forcing per $N_d$ perturbation, represented as follows:

$$F_0 = \frac{dSW_{\mathrm{TOA_{up}}}}{dln(N_d)} = \frac{dA_c}{dln(N_d)} \cdot f_c \cdot SW_{\mathrm{TOA_{dn}}}[\mathrm{Wm}^{-2}\mathrm{ln}(\mathrm{N_d})^{-1}].$$  (4)

Similar forms of this representation of forcing per perturbation have been used in, for example, Douglas and L'Ecuyer (2019) and Painemal (2018).

Uncertainties embedded in sensors' measuring precision and retrieval techniques/algorithms have been studied and well understood, and hence minimized in this study by choosing the right sensing channel and the rather strict quality control thresholds for cloud property retrievals (see Section 2 for details). However, uncertainties related to the methodology, that is linear regression errors of the slopes ($\beta_1$) of the $A_c$–$N_d$ relationship, are left to be quantified. A least-squares linear regression takes the form of:

$\hat{y} = \beta_0 + \beta_1 \cdot x$  (5)





where $\hat{y}$ is the estimated dependent variable of the linear model, $\beta_0$ is the intercept parameter and $\beta_1$ is the slope parameter. According to Press et al. (1988), the standard error of the slope parameter ($S_{\beta_1}$) can be expressed as:

$$S_{\beta_1} = \sqrt{\frac{SSE/(n-2)}{S_{xx}}} \qquad (6)$$

where $SSE$ is the residual sum of squares, which takes the form of:

$\qquad SSE = \Sigma(y_i - \hat{y}_i)^2 = \Sigma(y_i - (\beta_0 + \beta_1 x))^2 \qquad (7)$

$n$ is the number of data points, or degree of freedom, in the linear model, and $S_{xx}$ is the measure of total amount of variation in the independent variable, $x$, which takes the form of:

$$S_{xx} = \Sigma(x_i - \bar{x})^2. \qquad (8)$$

To construct confidence intervals around the calculated slope parameter, we use a t-distribution with $n-2$ degrees of freedom,
implied from the assumptions of the a simple linear regression model (Montgomery and Runger, 2010). As a result, the range of the regressed slopes takes the form of $\beta_1 \pm t_{\alpha/2, n-2} \cdot S_{\beta_1}$, where $100(1-\alpha)\%$ indicates the confidence interval. Accordingly, we report the 95% ($\alpha = 0.05$) confidence interval for our regressed slopes that characterize the $A_c$–$N_d$ relationship. Note $t_{0.025, n-2} \approx 2$ for $n - 2 \geq 6$.

    As we care about how cloud albedo susceptibilities vary with changing cloud states, meteorological conditions, and aerosol
loadings, both properties representing cloud states, e.g. LWP and $N_d$, and the ERA5 meteorological variables during the Aqua overpass over a 2-hour period are averaged to a $1°$ grid, in order to be associated with the calculated cloud albedo susceptibilities in the same space-time frame. As we are interested in averaged cloud properties within the $1°$ grid rather than cloudy scene properties, properties of cloudy CERES-MODIS footprints are averaged and weighted by their cloud fraction to obtain $1°$–mean cloud properties (Minnis et al., 2011a); therefore, overcast footprints are weighted heavily over partially
cloudy footprints. Because $N_d$ is only calculated when the footprint is overcast, only overcast footprints that also meet the rest of the $N_d$ retrieval criteria are used for the $1°$ $N_d$ averaging. To be consistent with $N_d$ averages, the rest of the $1°$–mean cloud properties are carried out the same way under the same conditions. This means our $1°$–mean cloud properties are represented by the average of only overcast footprints inside the grid with equal weights, as the cloud fractions of these overcast footprints are all equal to 1. Moreover, the same conditions under which the $S_0$ calculations are carried out are also applied to the $1°$
averaging, to avoid $1°$ scenes that lack overcast footprints contaminated by clouds that are not in the liquid phase, or possess multiple layers.

## 4 Results

### 4.1 Mean-state $A_c$-LWP-$N_d$ relationship

The mean-state $A_c$-LWP-$N_d$ relationship of the marine low-clouds over the northeast Pacific is shown as an average using
equally sized $N_d$ bins (10 cm$^{-3}$). Note that a relationship deduced from equally sized $N_d$ bins removes the dependence of





the relationship on the $N_d$ distribution, resulting in clearer physical relationships among these properties that are less affected by anthropogenic activities that can cause shifts in the $N_d$ distribution (Gryspeerdt et al., 2017, 2019a). Moreover, the cloud albedos used in this analysis are adjusted to a constant solar zenith angle (SZA) of $0°$, such that the dependence of $A_c$ on the seasonally varying SZA is removed, in order to obtain cleaner $A_c$-LWP-$N_d$ relationships. This is done based on the two-stream
approximation (Meador and Weaver, 1980) with the scattering asymmetry parameter approximated as a constant of 0.85 (Sagan and Pollack, 1967; Hu and Stamnes, 1993), using CERES-MODIS measured SZA and retrieved $\tau$.

From a climatological mean-state perspective, precipitating stratocumulus (Sc; approximated by $r_e$ greater than 12 $\mu m$ at cloud top for $c_w = 2.14 \times 10^6$ kg m$^{-4}$) become brighter as $N_d$ increases (Fig. 1, blue dots). This can be attributed, in part, to the increasing liquid water path (LWP; Fig. 1, black dots), consistent with the cloud lifetime effect (Albrecht, 1989), a
macrophysical effect on $A_c$. However, the increase in $A_c$ with increasing $N_d$ does not stop after the LWP reaches a plateau of $\sim$120 g m$^{-2}$ (at $N_d \approx 20$ cm$^{-3}$), suggesting a decrease in cloud effective radius ($r_e$) that contributes to the brightening of the cloud field, a microphysical effect on $A_c$ (Twomey, 1974, 1977). $A_c$ reaches a plateau of $\sim$0.32 (at $N_d \approx 100$ cm$^{-3}$) when Sc transitions into the non-precipitating regime ($r_e \leq 12$ $\mu m$) where negative LWP adjustments to increasing $N_d$ start to play a dominant role in changes in $A_c$.

For non-precipitating Sc, LWP decreases with increasing $N_d$, more markedly when the evaporation-entrainment feedback (EEF; Wang et al., 2003; Xue and Feingold, 2006) becomes more active (right hand side of the EEF isoline on Fig. 1). The strong EEF process that drives a dramatic decrease in LWP (dln(LWP)/dln($N_d$) $\sim$ -0.81) leads to a reduction in $A_c$ with increasing $N_d$ until LWP drops below $\sim$ 55 g m$^{-2}$ (Fig. 1, red circular outlines), after which $A_c$ increases with $N_d$ despite a continuous reduction in LWP, although more than halved in slope (dln(LWP)/dln($N_d$) $\sim$ -0.38) compared to the state when LWP
is above 55 g m$^{-2}$. This increase in $A_c$ with increasing $N_d$ after LWP drops below 55 g m$^{-2}$ can be explained by a decrease in entrainment efficiency as LWP decreases (Hoffmann et al., 2020) and an enhanced Twomey effect for less reflective thin clouds (Platnick and Twomey, 1994). The framework for discussion is the commonly used approximation of cloud albedo response to aerosol perturbations (e.g. Bellouin et al., 2020),

$$S_0 \;=\; \frac{dln(A_c)}{dln(N_d)} = \frac{1-A_c}{3}\left(1+\frac{5}{2}\frac{dln(LWP)}{dln(N_d)}\right) \tag{9}$$

in which dln(LWP)/dln($N_d$) of -0.4 marks the critical value of the LWP adjustment in the entrainment/non-precipitating regime, as it determines the overall sign of the albedo susceptibility approximation, i.e. a warming (negative) or a cooling (positive) effect (e.g. Glassmeier et al., 2021).

The climatological mean-state indicates an overall positive response of $A_c$ to $N_d$ perturbations (a cooling effect), despite an overall negative LWP adjustment (dln(LWP)/dln($N_d$) $\sim$ -0.58) that would be sufficient to overcome the Twomey effect and
lead to warming, for these high $f_c$ non-precipitating Sc over the NE Pacific region (Fig. 1). The strong and sufficiently negative LWP adjustment derived in this study from long-term satellite observations is in agreement with assessment of Glassmeier et al. (2021) for the same region and regime (a lower bound dln(LWP)/dln($N_d$) = -0.64), but based on an ensemble of large-eddy simulations. Such agreement between the results learned from an ensemble of model simulated time-evolving nocturnal stratocumulus systems and results deduced from a large composite of remote satellite sensors captured afternoon stratocumulus





properties might suggest a robustness of these characteristics regarding the relationship between $A_c$, $N_d$ and LWP of marine stratocumulus. The result from this work, in addition, points to the importance and necessity of considering the more strongly entraining regime of thicker clouds (LWP > 55 g m$^{-2}$) and the weakly entraining while strongly Twomey-brightening regime of thinner clouds (LWP < 55 g m$^{-2}$) separately; the strength of LWP adjustment is more than halved in the latter ($\sim$ -0.38) compared to the former regime ($\sim$ -0.81), allowing the Twomey effect brightening to prevail.

## 225  4.2  Albedo susceptibility in a LWP-$N_d$ space and susceptibility regimes

Cloud albedo susceptibility is displayed in the LWP-$N_d$ space, with the size of the circles indicating the frequency of occurrence of a particular cloud state (Fig. 2). Precipitating Sc ($r_e > 12$ $\mu m$) present an overall cloud brightening potential per $N_d$ perturbation, indicated by the mostly positive susceptibilities, except for some LWP-$N_d$ states that are in the entrainment-evaporation regime (left of the $r_e$=12 $\mu m$ isoline and right of the EEF isoline on Fig. 2). An occurrence-weighted mean

radiative susceptibility ($F_0$) of 10.5 $\pm$ 1.45 W m$^{-2}$ ln($N_d$)$^{-1}$ corresponding to the precipitating Sc with positive $S_o$ reflects the role of the cloud lifetime effect (Albrecht, 1989, and Fig. 1), such that increases in $N_d$ suppress the warm rain process, favoring the development of deeper and brighter clouds. This regime is hereafer referred to as *the precipitating-brightening regime*. It occurs $\sim$22% of the time when single-layer liquid cloud is present over the NE Pacific, based on this 10-year satellite-derived climatology.

For non-precipitating Sc, two regimes emerge in the LWP-$N_d$ space, indicated by the changing sign of albedo susceptibility at LWP $\approx$ 55 g m$^{-2}$, with thicker Sc (LWP > 55 g m$^{-2}$) showing a cloud darkening potential (negative $S_0$) and thinner Sc (LWP < 55 g m$^{-2}$) showing a strong cloud brightening potential (positive $S_0$) per $N_d$ perturbation (Fig. 2). This is consistent with the "inverted V-shape" dependence of mean-state $A_c$ as a function of $N_d$ for non-precipitating Sc shown in Fig. 1 (blue dots), with the turning point being around 55 g m$^{-2}$. As discussed in Section 4, the non-precipitating cloud states with negative $S_0$

are dominated by the entrainment driven LWP adjustment ($\sim$-0.81, Fig. 1 brown fitting line) which is double the critical slope value (-0.4) for entering the warming regime (Glassmeier et al., 2021). This entrainment-evaporation regime cloud state (right of the EEF isoline on Fig. 2) with negative $S_0$ occurs $\sim$36% of the time when single-layer liquid cloud is present. It produces an occurrence-weighted $F_0$ = -20.2 $\pm$ 1.86 W m$^{-2}$ ln($N_d$)$^{-1}$, and is hereafter referred to as *the entrainment-darkening regime* (mostly non-precipitating).

The thinner Sc (LWP < 55 g m$^{-2}$) not only possess strong positive albedo susceptibilities for reasons discussed in Section 4, but these cloud states also occur the most frequently ($\sim$37% of the time when single-layer liquid cloud is present; Fig. 2). As a result, a dominating positive occurrence-weighted mean $F_0$ of 30.7 $\pm$ 1.55 W m$^{-2}$ ln($N_d$)$^{-1}$ is associated with these non-precipitating cloud states with positive $S_0$, hereafter referred to as *the Twomey-brightening (non-precipitating) regime*. Climatologically, the cloud-state dependent albedo susceptibilities and their corresponding frequency of occurrence together

determine that the stratocumulus deck over the NE Pacific presents an overall cloud brightening potential with an occurrence-weighted $F_0$ of 20.8 $\pm$ 0.96 W m$^{-2}$ ln($N_d$)$^{-1}$ (Fig. 2), in agreement with the results shown in Fig. 1.



### 4.3 Meteorological constraints

One of the main questions we want to address is under what meteorological conditions are marine low-clouds most/least susceptible to aerosol perturbations, or in other words, what is the influence of meteorology on albedo and radiative suscepti-
bilities? Then, by quantifying the frequency of occurrence of susceptible conditions, and the potential radiative effect associated therewith, we have the means to quantify the radiative effect of aerosol-cloud interactions. In this section, we assess meteorological constraints on low-cloud albedo susceptibility from multiple perspectives, including a mutual information analysis (4.3.1), where to find susceptible and less susceptible conditions in meteorological factor spaces (4.3.2), the role of seasonal covariability in meteorological conditions (4.3.3), and the impact of individual meteorological factors on the occurrence of
susceptibility regimes and the overall occurrence-weighted radiative susceptibility (4.3.4).

### 4.3.1   Mutual information analyses reveal primary governance of LWP, $N_d$ and CTH on $S_0$

First, we quantify how much information, treated as entropy (Shannon, 1948), is shared between individual meteorological factors (MFs) and albedo susceptibilities, using a statistical technique called mutual information (MI) analysis (Fig. 3). We follow the methodology in Glenn et al. (2020). Because MI analysis doesn't require a pre-defined relational function between
variables, it handles nonlinear relationships, which is the case for this study (i.e. albedo susceptibility and meteorological factors), just as well as linear relationships. Cloud top heights of marine stratocumulus, marine boundary layer heights, and inversion heights are positively correlated in the setting of the stratocumulus-topped boundary layer over the NE Pacific. Therefore, CTH is considered here as a variable governing the cloud states while reflecting a meteorological condition at the same time.

Although the percentage of shared information between $S_0$ and meteorological conditions remains very low (less than a percent) for all factors investigated in this study, the MI analysis reveals a leading role of cloud top height (~1%) in terms of covariability with $S_0$, whereas the MI of all other factors are comparable to each other (between 0.1% to 0.3%), with boundary layer (BL) meridional winds and $RH_{ft}$ being the second to highest (~0.3%; Fig. 4a). The leading role of CTH is consistent with the clear separation between entrainment-darkening and Twomey-brightening regimes in LWP-$N_d$ space (Fig. 2), and
the secondary role of BL meridional winds can be explained by the fact that relatively polluted continental flows (northerlies) advect aerosol to our study area (120–130°W, 20–30°N), whereas southerly flows of a oceanic origin tend to advect cleaner air. The MI between $S_0$ and the zonal component of the boundary layer wind is half of that with the meridional component (not shown), suggesting meridional winds are more tightly connected to continental/oceanic flows and thereby variations in aerosol loading and $N_d$ in our study area.

Next, we examine the unique information contained in individual MFs, if some variable representing a particular cloud state, e.g. LWP, $N_d$, or CTH, is known, using the method called conditional MI (CMI) analysis, also following Glenn et al. (2020). When the MI analysis is conditioned on $N_d$, LWP, and CTH, the percentage of shared information between $S_0$ and MFs increases by almost a factor of 10 (Fig. 3b-d), meaning the amount of unique information about $S_0$ contained in LWP, $N_d$, and CTH is almost a factor of 10 greater than those contained in individual MFs. Moreover, we repeat the CMI analysis between the





same set of MFs and a randomly permuted $S_0$ sample space (representing noise; reported as noise-CMI), in order to estimate
the baseline signal of these CMIs, by taking the difference between the CMIs and noise-CMIs (Fig. 3, light gray bars). The
baseline signal strength suggests that if LWP or $N_d$ or CTH is known, the unique information remaining in individual MFs
that is shared with $S_0$ is less than a percent different from that which is shared with noise. When one conditions on $N_d$, the
secondary role of the BL meridional winds is no longer evident, and all MFs beside CTH have almost the same CMI, consistent
with the idea of lower-level wind driving the variability in $N_d$. When one conditions on LWP, the leading role of CTH is much
reduced, as CTH correlates with LWP, especially for non-precipitating Sc. Worth noting is that CTH still possesses the highest
CMI among all MFs, although the lead margin is much reduced compared to the unconditioned case, suggesting other critical
roles of CTH under constant LWP, such as the cloud top entrainment drying feedbacks. Last but not least, when conditioning
on CTH, all other MFs have very similar CMIs of about 2%.

295        From the MI and CMI analyses, we conclude that meteorological conditions affect the albedo susceptibility of low-clouds
mainly through governing the states of the clouds, i.e. LWP, $N_d$ and CTH. If these cloud state indicators are known or pri-
defined, e.g. for a given cloud state (LWP, $N_d$, CTH), meteorological conditions associated with that state share very little
information with the $S_0$ of those clouds. This is consistent with the concept of "equifinality" (von Bertalanffy, 1950; Mülmen-
städt and Feingold, 2018), where many different meteorological conditions can yield the same state (LWP, $N_d$, CTH), thereby
obscuring unique matchings between meteorological conditions and $S_0$, and resulting in overall low MI between MFs and $S_0$.

### 4.3.2    Susceptible and less susceptible conditions in meteorology spaces

We map cloud states in the LWP-$N_d$ space (Fig. 2) directly onto meteorological spaces (Fig. 4), to reveal the association
between meteorological conditions and the radiative susceptibility regimes identified in Section 4.2. A clear separation of the
entrainment-darkening and Twomey-brightening regimes is evident in all 6 meteorological spaces (Fig. 4, brown and green/blue
open circles), more markedly in the direction of cloud top height (Fig. 4a-c). Moreover, these 2 regimes tend to cluster in
meteorological spaces: the Twomey-brightening regime clusters at low CTH, highest LTS, relatively low SST, and lowest
$RH_{ft}$, and the entrainment-darkening regime clusters at higher CTH, lower LTS, higher SST, and higher $RH_{ft}$, compared to
the Twomey-brightening regime (Fig. 4). The clustering of these two regimes in these meteorological spaces is consistent with
their states in the LWP-$N_d$ space, as stratocumulus with higher cloud tops usually have higher LWP over the NE Pacific region.
Therefore, thicker and deeper clouds are more strongly affected by the cloud-top entrainment feedbacks, leading to decreases
in LWP as $N_d$ increases, whereas thinner and lower Sc are subject to less effective entrainment processes, maintaining the cloud
LWP such that an increase in $N_d$ can sufficiently decrease $r_e$ and brighten the clouds. The vertical extent of the subtropical
marine stratocumulus or the depth of the stratocumulus-topped boundary layer (STBL) is controlled, to first order, by the
LTS at longer time scales (Eastman et al., 2017) and $RH_{ft}$ at shorter time scales (Eastman et al., 2017; Eastman and Wood,
2018), such that enhanced LTS (a stronger buoyancy gradient across the inversion) or higher free-tropospheric humidity (less
radiative and evaporative cooling), all else being equal, limits the entrainment of free tropospheric air and thereby suppresses
the deepening of marine boundary layers. Hence, the primary occurrence of the Twomey-brightening regime is under the
highest LTS conditions, however, perhaps counterintuitively, also under the lowest $RH_{ft}$ conditions (Fig. 4b, e, and f). This is





because large-scale meteorological conditions are strongly correlated over eastern subtropical oceans where the Earth's major
marine stratocumulus decks are formed (Wood, 2012), such that LTS and $RH_{ft}$ are negatively correlated (evident in Fig. 4e and
further discussed in 4.3.3), as prevailing free-tropospheric subsidence transports dry upper-level air downward and increases
the stability.

In contrast, the precipitating-brightening regime tends to spread out in the meteorological spaces, overlapping with the
other two regimes, except in the spaces of $RH_{ft}$ and LTS (e.g. Fig. 4e). This suggests precipitation-suppression driven cloud
brightening tends to occur, first, when LTS is weak (less than 21 K), regardless of $RH_{ft}$ or SST; second, when the free-
troposphere is the moistest (> 45%) co-occurring with the highest SST conditions (> 294.5 K) (Fig. 4f). Despite high SST
conditions, the precipitating-brightening branch appears under high $RH_{ft}$, suggesting a dominant role of the free-tropospheric
humidity. Here, enhanced free-tropospheric humidity (a reduced humidity gradient across the cloud top) slows/weakens droplet
evaporation, creating favorable conditions for precipitation, which is susceptible to aerosol induced warm-rain suppression
process, and thereby cloud brightening. This role of $RH_{ft}$ is reinforced by the fact that the precipitating-brightening branch is
displaced from the non-precipitating branch in Fig. 4f, where $RH_{ft}$ alone determines which susceptibility regimes the clouds
will be in at a constant SST.

### 4.3.3   The role of seasonal covariability in meteorological conditions

Monthly climatologies of ERA5 meteorological factors, including LTS, SST, $RH_{ft}$, and 700 hPa subsidence, averaged over the
NE Pacific show a strong seasonality and a tight correlation among these factors (Fig. 5a). The annual cycle in SST (blue) and
700 hPa vertical velocity (gray) are correlated and anti-correlated with that of the Northern Hemispheric insolation, respectively
(not shown), such that summer time (June–September) SST is the highest whereas free-tropospheric subsidence is the weakest
due to a weakened Hadley circulation when insolation is at its annual maximum in the Northern Hemisphere. Moreover, the
annual cycle in free-tropospheric humidity (black) is very well anti-correlated with that of the free-tropospheric subsidence,
leading to a positive (although lagged) correlation between $RH_{ft}$ and SST (also evident in Fig. 4f). As the Hadley circulation
starts to strengthen in January, indicated by the enhancing 700 hPa subsidence (January to May), and SST over the subtropical
ocean remains cool during boreal spring, LTS (red) increases markedly. SST starts to increase as the Northern Hemisphere
enters its summer season, resulting in a weakening of the Hadley circulation and the free-tropospheric subsidence, and leading
to a continuous decrease in LTS from June until January. As a result, LTS peaks in June, leading the annual maximum in SST
by 3 months (Fig. 5a).

In response to the strengthening LTS during boreal spring, both CTH (black) and cloud LWP (blue) decrease, with cloud
LWP reaching its annual minimum in May (Fig. 5b). The thinnest clouds of the year give rise to the annual maximum in the
occurrence of the Twomey-brightening regime in May, resulting in an annual maximum of $F_0$ (Fig. 5c). As LTS decreases and
SST continues to warm during boreal summer and fall, cloud LWP and CTH increase until December, when LTS is at its annual
minimum and the precipitating-brightening regime is at its annual maximum occurrence, resulting in a secondary peak in the
annual cycle of $F_0$. During the boreal summer months (June–September), when SST is the highest, the entrainment-darkening
regime is at its annual maximum occurrence, resulting in the lowest $F_0$ throughout the annual cycle. The high summertime $N_d$





also favors the occurrence of the entrainment-darkening regime through the entrainment feedbacks. This is in agreement with the finding that warmer SST over the northeast (NE) Atlantic leads to mostly darkening clouds (Zhou et al., 2021). Although $F_0$

responds to SST over the NE Pacific the same way as it does over the NE Atlantic, marine low-clouds over the NE Pacific never enter an overall darkening regime, likely due to the co-occurring high free-tropospheric humidity and high SST conditions and thereby a relatively persistent and high occurrence of the precipitating-brightening regime (July–September), which is rarely the case for the high SST conditions over the NE Atlantic in Zhou et al. (2021).

### 4.3.4 Meteorology affects the occurrence of albedo susceptibility regimes

As discussed in Section 4.3.1, meteorological or environmental conditions influence the albedo susceptibility of a cloud field to aerosol perturbations through regulating the states of the clouds, e.g. their $N_d$, LWP and CTH. Just as important as the role of seasonal covariability in MFs on cloud albedo susceptibility is the role of individual MFs, which was obscured by the monthly evolution in meteorological conditions in Section 4.3.3. Hence, in this section, we further examine the occurrence and the strength of each albedo susceptibility regime identified in the LWP-$N_d$ space (Section 4.2), as a function of individual MFs

(Figs. 6-8).

#### a. Cloud top height (CTH)

As cloud top heights of marine Sc increase or as the Sc-topped boundary layers deepen, clouds are more likely to develop

higher LWPs and are more likely to precipitate. A pronounced decrease in occurrence-weighted radiative susceptibility with increasing CTH, from 60.5 W m$^{-2}$ ln($N_d$)$^{-1}$ to -40.3 W m$^{-2}$ ln($N_d$)$^{-1}$, is noted (Fig. 6a). We choose not to report the uncertainties associated with these $F_0$s as they are all of the same order and rather redundant to the quantitative comparison discussed in this section. The remarkable decrease in $F_0$ can be reasoned through two contributing mechanisms, i) changes in the magnitude of $S_0$ and ii) a shift in the frequency of occurrence of cloud states (LWP, $N_d$), as cloud top elevates. First,

regarding changes in the magnitude of $S_0$, a clear enhancement in the negative susceptibilities in the entrainment-darkening regime, by -0.16, is evident as CTH increases (Fig. 6a and dashed curves in 6c), consistent with an increasing influence of the entrainment feedbacks as cloud deepens. For the precipitating-brightening Sc, $S_0$ decreases slightly with increasing CTH, by - 0.05, leading to a steady decrease in regime-mean $F_0$, by -10.1 W m$^{-2}$ ln($N_d$)$^{-1}$ (Fig. 6b), given little change in the occurrence of the regime. This could reflect two possible balancing mechanisms: i) a balance between warm rain suppression and the

increasing precipitation (droplet removal) efficiency with deeper/higher clouds; ii) a balance between warm rain suppression and the strengthening entrainment drying with higher cloud tops.

Second, a pronounced shift in the occurrence of the albedo susceptibility regimes (Fig. 6a and solid curves in 6c) is perhaps more evident, such that the marine Sc over the NE Pacific are more likely to be found in the entrainment-darkening regime (55%) rather than the Twomey-brightening regime (11%) in the highest CTH quartile. This is in contrast to the lowest CTH

quartile, where the Twomey-brightening regime (55%) is much more likely to occur than the entrainment-darkening regime (14%). This shift in regime occurrence (and the MFs that define them) as CTH increases is the primary driver of the significant



changes in the overall occurrence-weighted $F_0$, in which the contribution from the Twomey-brightening regime shrinks by 44.6 W m$^{-2}$ ln($N_d$)$^{-1}$, and the contribution from the entrainment-darkening regime increases by 55.5 W m$^{-2}$ ln($N_d$)$^{-1}$ (Fig. 6b).

**b. Lower-tropospheric stability (LTS)**

Given the fact that LTS and RH$_{ft}$ are negatively correlated over subtropical marine stratocumulus regions (Fig. 5), data are examined in 6 equally populated LTS-RH$_{ft}$ bins according to their joint histogram (Fig. 7a-f). As expected, the bin with highest LTS (greater than 25 K) is associated with the lowest RH$_{ft}$ (bin-mean of 17 %) (Fig. 7a). The $F_0$ associated with this
condition is the highest (36.1 W m$^{-2}$ ln($N_d$)$^{-1}$) among the 6 LTS-RH$_{ft}$ bins, mainly owing to the high occurrence of the Twomey-brightening regime (occurring 50% of the time; Fig. 7a), contributing an $F_0$ of 47 W m$^{-2}$ ln($N_d$)$^{-1}$. If we stay in this relatively dry free-troposphere and reduce LTS, $F_0$ decreases from 36.1 to 23.7 and to 9.6 W m$^{-2}$ ln($N_d$)$^{-1}$ (Fig. 7a, b, d). In such a case, unstable conditions facilitate growth of cloud LWP. Deepening is associated with entrainment, which shifts more clouds away from the Twomey-brightening regime into the other two regimes (Fig. 7d), leading to stronger entrainment-
darkening $F_0$, from -17.3 to -24.3 W m$^{-2}$ ln($N_d$)$^{-1}$, and weaker Twomey-brightening $F_0$, from 47 to 27.1 W m$^{-2}$ ln($N_d$)$^{-1}$. Although the occurrence of the precipitating-brightening regime in the lowest LTS bin (Fig. 7d) is double that of the highest LTS bin (Fig. 7a), the high cloud-tops associated with the lowest LTS limit the cloud brightening potential from warm rain suppression, similar to discussions related to Fig. 6d, through a balance between rain suppression and droplet removal efficiency (via precipitation) and/or a balance between rain suppression and entrainment drying.
If one simply composites the data as a function of LTS alone, knowing that RH$_{ft}$ will change accordingly to LTS, increasing LTS mostly affects the occurrence of the precipitating-brightening regime (by -25%) and the Twomey-brightening regime (by +24%) (Fig. 8b), leading to changes in the occurrence-weighted regime-$F_0$ of -11.4 and +30.6 W m$^{-2}$ ln($N_d$)$^{-1}$, respectively (Fig. 8a). The summation of the 3 regime-$F_0$s results in an overall increase in $F_0$ by ∼20 W m$^{-2}$ ln($N_d$)$^{-1}$.

**c. Free-tropospheric humidity (RH$_{ft}$)**

The effect of RH$_{ft}$ on radiative susceptibility under similar stability conditions has two aspects. First, moister air above cloud tops reduces the humidity gradient across the cloud-top inversion thereby weakening the evaporation-entrainment feedback. This is evident in comparisons between Fig. 7b-c and between 7d-f, where the darkening potential (negative $S_0$, brown circles)
is reduced under higher RH$_{ft}$ conditions, more markedly at the highest RH$_{ft}$ (Fig. 7f). Second, as conditions in the free-troposphere become more humid, marine low-level clouds are more likely to possess higher LWP and reside in a more favorable environment for precipitation, indicated by the high occurrence of the precipitating-brightening regime (42%) in Fig. 7f. ERA5 humidity profiles also indicate a positive correlation between RH$_{ft}$ and the RH within the boundary layer (not shown), further supporting higher LWP. The increase in LWP with increasing RH$_{ft}$ leads to a shift in cloud state away from the Twomey-
brightening regime, towards the other two regimes, but mostly towards the precipitating-brightening regime (Fig. 7f and 8d).



Worth noting is that the magnitude of these two effects of $RH_{ft}$ on albedo susceptibility and their occurrence amplify as $RH_{ft}$ increases (the steep changes at the highest 20 percentile of $RH_{ft}$ in Fig. 8d).

Overall, the Twomey-brightening regime is the regime most sensitive to variations in $RH_{ft}$ and LTS (Fig. 8), which are often controlled by the large scale vertical motion in the free-tropospheric. The sensitivity is mainly reflected in the frequency-

of-occurrence. High cloud brightening potential is associated with either the highest LTS (36.1 W m$^{-2}$ ln(N$_d$)$^{-1}$; Fig. 7a) co-occurring with the lowest $RH_{ft}$, i.e., conditions favoring Twomey-brightening, or the highest $RH_{ft}$ (25.2 W m$^{-2}$ ln(N$_d$)$^{-1}$; Fig. 7f), co-occurring with the lowest LTS, i.e., conditions favoring precipitating-brightening.

### d. Sea surface temperature (SST)


As sea surface temperature increases over the NE Pacific, radiative susceptibility decreases, from 39.9 to 6.1 W m$^{-2}$ ln(N$_d$)$^{-1}$ (Fig. 9a). First, SST changes are the driver of changes in many other meteorological factors, e.g. surface fluxes, MBL height, LTS, and humidity. Here, we do not attempt to separate out the role of SST on radiative susceptibilities while controlling for other MFs, but rather explore the radiative susceptibility as a function of SST, with all the inherent covariability

between SST and other MFs. In general, cloud states shift towards higher LWP and lower N$_d$, an indication of thicker clouds with larger droplet sizes, as SST increases, suggesting a higher likelihood of precipitation and scavenging for the clouds in the warmer SST conditions (more circles to the left of the 12 $\mu m$ isoline on Fig. 9a rightmost panel). This is consistent with an increase in SST leading to an increase in surface fluxes and a weaker LTS in a well-mixed marine boundary layer, both supporting the development of deeper Sc with higher LWPs (similar to the response of trade-wind cumulus to warming in

Vogel et al. (2016)). Another effect associated with thicker clouds is the creation of favorable conditions for the entrainment feedbacks, which is shown as a strengthening of the entrainment-darkening S$_0$ (Fig. 9a, brown circles getting darker). As a result, as SST increases, the increasing occurrence of the strengthening entrainment-darkening regime and the decreasing occurrence of the Twomey-brightening regime (Fig. 9c) lead to the overall decrease in F$_0$, by ~34 W m$^{-2}$ ln(N$_d$)$^{-1}$ (Fig. 9a, leftmost vs rightmost).

In the current climate, the free-tropospheric humidity over the NE Pacific correlates well with SST through the seasonality in large-scale circulation (i.e. the free-tropospheric subsidence related to the Hadley circulation), such that higher SST is associated with enhanced above-cloud humidity, favoring the occurrence of the precipitating-brightening regime (Fig. 9c, the "U" shaped occurrence variation of the precipitating-brightening regime). The rebounding of the precipitating-brightening regime at high SST conditions (Fig. 9b and c) partially offsets the darkening potential that would otherwise dominate the overall

radiative susceptibility, leading to a warming effect, in the absence of the enhanced free-tropospheric humidity. However, if SST continues to rise in the coming decades, assuming the same trend observed in Fig. 9, we might expect the NE Pacific stratocumulus region to exhibit an overall darkening potential to aerosol perturbations.

In the assessment of the role of individual MFs, we do acknowledge that a change in one MF can be associated with changes in other MFs (the seasonal covariability in meteorological conditions as an example). Our goal has been to retain this

covariability between MFs in our analyses, as we aim to quantify influences of meteorology on radiative susceptibility in the





manner in which nature is observed. This is in contrast to a traditional investigation of individual MFs when all others are held constant. The latter approach only represents a small portion of the natural variability, and the role of covariabilities between MFs is missed.

## 5 Concluding remarks

This study quantifies the albedo susceptibility and radiative susceptibility to $N_d$ perturbations of high $f_c$ single layer, marine low-clouds over the NE Pacific stratocumulus region, using 10 years of MODIS-retrieved daytime cloud properties and CERES-measured radiative fluxes at the top-of-atmosphere. A novel aspect of this study is the assessment of susceptibility across a LWP-$N_d$ space, such that albedo susceptibility associated with individual cloud states (LWP, $N_d$) and, more importantly, their frequencies of occurrence are quantified. Moreover, the effects of ERA5 meteorological factors and their covari-

ability, on the albedo susceptibility are explored. This allows us to quantify conditions under which low-clouds are most/least susceptible to aerosol perturbations, and how frequently these conditions occur. Robust establishment of three albedo susceptibility regimes is found regardless of meteorological states or environmental conditions, however, the occurrence and strength of these regimes are clearly modified by meteorological conditions. Key findings are:

1. From a climatological mean-state perspective, LWP and $N_d$ are negatively correlated for non-precipitating Sc (Fig.
1), consistent with previous polar-orbiting satellite based studies (e.g. Gryspeerdt et al., 2019a; Possner et al., 2020). Results from the current study, however, indicate that despite the negative LWP adjustment, cloud albedo increases with increasing $N_d$ for non-precipitating Sc overall, pointing to the importance of considering the high-LWP cloud states separately from the low-LWP cloud states, as the negative LWP adjustments are clearly different for thicker versus thinner Sc (Fig. 1).

2. When cloud albedo susceptibility is mapped onto a LWP-$N_d$ state space, three susceptibility regimes emerge: i) the Twomey-brightening regime (occurring 37%, contributing $30.7 \pm 1.55$ W m$^{-2}$ ln($N_d$)$^{-1}$), consisting of non-precipitating thinner clouds (LWP < ~55 g m$^{-2}$) and consistent with a dominating Twomey effect for clouds of relatively low albedo and weaker entrainment; ii) the entrainment-darkening regime (occurring 36%, contributing $-20.2 \pm 1.86$ W m$^{-2}$ ln($N_d$)$^{-1}$), comprising mostly non-precipitating thicker clouds (LWP > ~55 g m$^{-2}$) and consistent with entrain-
ment feedbacks that drive a decrease in LWP with increasing $N_d$; iii) the precipitating-brightening regime (occurring 22%, contributing $10.5 \pm 1.45$ W m$^{-2}$ ln($N_d$)$^{-1}$), comprising precipitating clouds with effective radii mostly greater than 12 $\mu m$ and consistent with the cloud lifetime effect due to a suppressed warm rain process (Fig. 2). An overall cloud brightening potential of $20.8 \pm 0.96$ W m$^{-2}$ ln($N_d$)$^{-1}$ is found for the marine low-clouds over the NE Pacific stratocumulus region, after the frequency of occurrence of each regime is accounted for.

3. Based on mutual information analysis, LWP, $N_d$ and CTH are shown to be the governing factors of low-cloud albedo susceptibility. Individual meteorological factors add very little (less than a percent) shared information with $S_0$, if the aforementioned three variables are known (Fig. 3). That said, meteorological factors are shown to affect the overall





radiative susceptibility of marine Sc but mainly through governing the frequency of occurrence of cloud states, i.e. LWP and $N_d$, and thereby the occurrence of each of the susceptibility regimes (Figs. 6-9).

4. When cloud states, along with their associated radiative susceptibilities, are mapped to meteorological spaces of LTS, SST, CTH, and $RH_{ft}$, the entrainment-darkening regime and the Twomey-brightening regime are clearly associated with distinct meteorological conditions. The Twomey-brightening regime occurs most frequently under low CTH, highest LTS, low SST, and lowest $RH_{ft}$ conditions. Such a combination of these meteorological factors occurs in May as a result of the seasonally covarying meteorological conditions related to the large-scale circulation over the NE Pacific.

The entrainment-darkening regime occurs most frequently under relatively high CTH and intermediate LTS, SST, $RH_{ft}$ conditions, which prevail during the boreal summer months (July–September). The precipitating-brightening regime mostly prefers unstable conditions (low LTS), occurring during winter months (November–January), but a very moist free troposphere (co-occurring with high SST in August) also promotes the occurrence of this regime (Figs. 4-5).

5. As cloud-top height or marine boundary layer height increases, cloud states shift towards larger LWP, resulting in a
pronounced decrease in the Twomey-brightening regime occurrence and a marked increase in the occurrence of the entrainment-darkening regime. This is accompanied by an enhanced entrainment-darkening susceptibility strength and a reduced precipitating-brightening susceptibility strength. As a result, $F_0$ decreases substantially with increasing CTH, from 60 to -40 W m$^{-2}$ ln($N_d$)$^{-1}$ (Fig. 6).

6. The influence of LTS on $F_0$ is mainly exerted via the occurrence of each susceptibility regime, rather than its mean $S_0$.
Strong stability (high LTS) leads to shallower Sc that mostly occur in the Twomey-brightening regime, whereas unstable conditions (low LTS) allow clouds to grow deeper and become more prone to precipitation, leading to high occurrence of the precipitating-brightening regime (Figs. 7-8).

7. A moist free-troposphere has two major impacts on the radiative susceptibility, i) a reduced humidity gradient across the cloud-top inversion weakens the evaporation-entrainment process, leading to a less negative LWP adjustment for
thicker non-precipitating clouds; ii) a moist free-troposphere gives rise to a higher occurrence of thicker and deeper clouds, driving a major shift of cloud states away from the Twomey-brightening regime, mostly into the precipitating-brightening regime (Figs. 7-8).

8. Increases in SST lead to a deeper marine boundary layer, lower LTS and thicker clouds. As a result, $F_0$ decreases with increasing SST, owing to a higher occurrence of deeper clouds (meaning less occurrence of the Twomey-brightening
regime) and a stronger entrainment-darkening regime associated with the weakened stability. In contrast to the NE Atlantic (Zhou et al., 2021), moist free-tropospheric conditions, co-occurring with high SSTs, during summertime over the NE Pacific, hamper the role of the strengthening entrainment-darkening regime, by shifting clouds towards the precipitating-brightening regime (Fig. 9).

By focusing on this marine stratocumulus dominated region/regime over the NE Pacific, we have robustly identified three
susceptibility regimes in the LWP-$N_d$ space and linked the responses to existing understanding of marine stratocumulus.

Future work quantifying the occurrence and strength of these three regimes at various oceanic locations, associated with different meteorological regimes/conditions, will enable an extended satellite-based assessment of the radiative susceptibility of global marine low-clouds. Moreover, if aerosol perturbations, natural or anthropogenic, are estimated in some form, the characterization and quantification of radiative susceptibility regimes provided in this study can be used to provide a global
estimate of radiative forcing or radiative effect, due to aerosol-marine low-cloud interactions. Such assessments are planned for a follow-on study.

*Data availability.* The CERES SSF data are publicly available from NASA's Langley Research Center (https://satcorps.larc.nasa.gov/). The fifth-generation ECMWF (ERA5) atmospheric reanalyses of the global climate data are available through the Copernicus Climate Change Service (C3S, https://cds.climate.copernicus.eu/).

*Author contributions.* JZ carried out the analysis and wrote the manuscript. XZ and GF contributed to developing the basic ideas, discussing the results, and editing the paper.

*Competing interests.* Graham Feingold is a co-editor of ACP. Other than this, the authors declare that they have no conflict of interests

*Acknowledgements.* Jianhao Zhang acknowledges support by a National Research Council Research Associateship award at the National Oceanic and Atmospheric Administration (NOAA).

*Financial support.* This research has been supported by the U.S. Department of Energy, Office of Science, Atmospheric System Research Program Interagency Award 89243020SSC000055 and the U.S. Department of Commerce, Earth's Radiation Budget grant, NOAA CPO Climate & CI #03-01-07-001. JZ was supported by the National Academies of Sciences, Engineering, Medicine (NASEM), National Research Council Research Associateship Award.



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



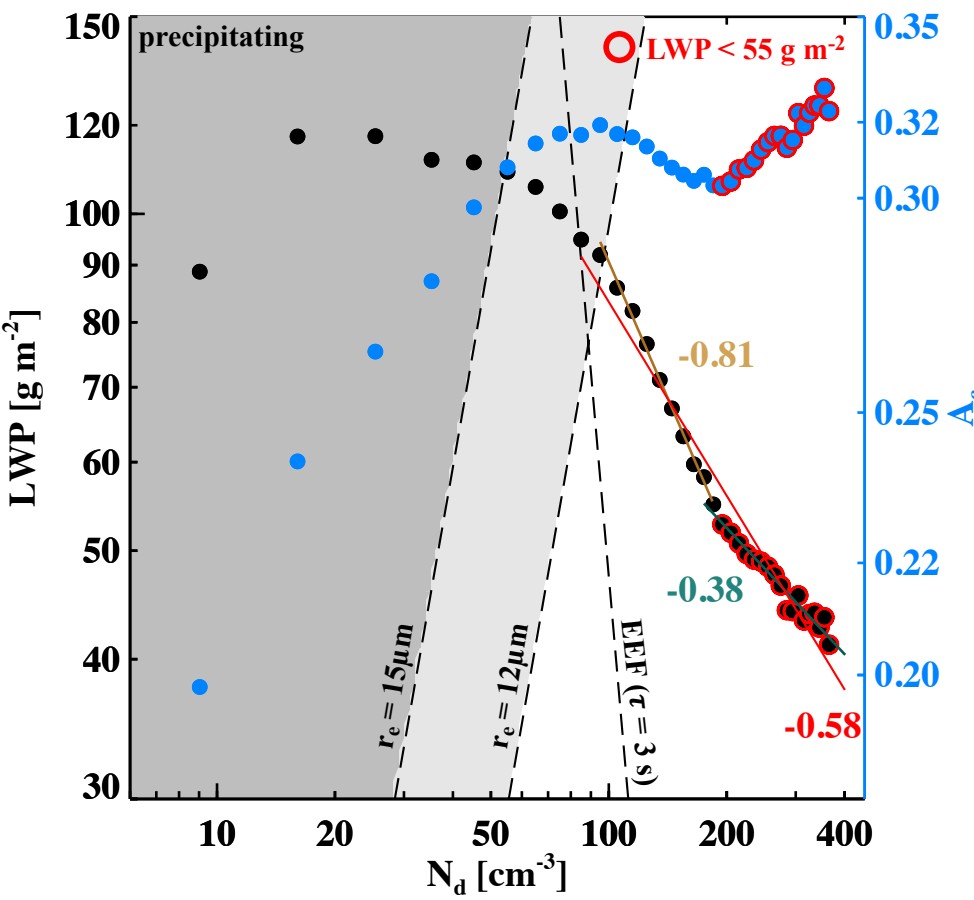

**Figure 1.** Mean liquid water path (LWP; black dots) and cloud albedo ($A_c$; blue dots) of each cloud droplet number concentration ($N_d$) bin (bin-size of 10 cm$^{-3}$). Values are shown on logarithm scales. Isolines of evaporation-entrainment feedback (EEF; phase relaxation timescale of 3 seconds), effective radius ($r_e$) of 12 $\mu m$ and 15 $\mu m$ (commonly used measures of precipitation) based on an adiabatic condensation rate of 2.14 x 10$^6$ kg m$^{-4}$, shades of grey background colors represent a general indicator of likelihood of precipitation, and bin-mean LWP less than 55 g m$^{-2}$ are highlighted by red circular outlines. The linear regressed slopes of ln(LWP)-ln($N_d$) for all non-precipitating clouds (red), non-precipitating clouds with LWP > 55 g m$^{-2}$ (brown), and non-precipitating clouds with LWP < 55 g m$^{-2}$ (green) are also indicated.

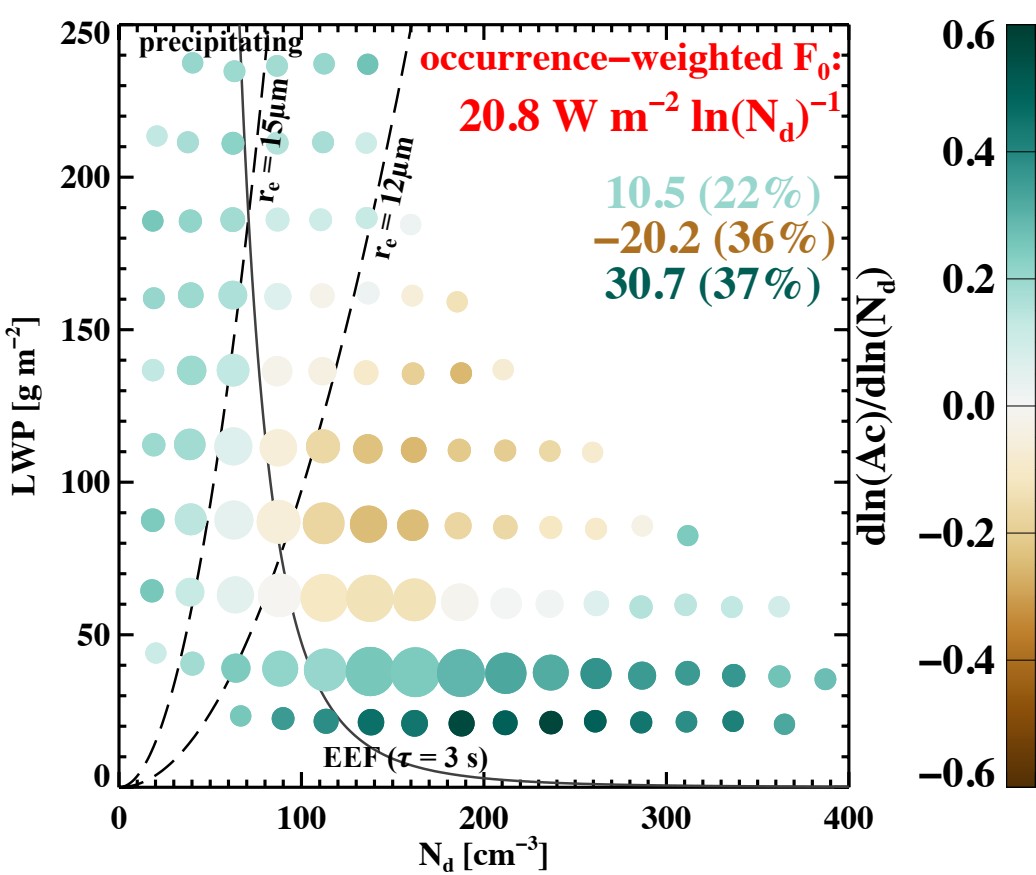

**Figure 2.** Cloud albedo susceptibility ($S_0$, colored filled circles) in LWP-$N_d$ space, as bin means (bin-size of 25 g m$^{-2}$ and 25 cm$^{-3}$). Isolines of $r_e$ of 12 $\mu m$ and 15 $\mu m$ (black dashed) and EEF (as in Fig. 1) are indicated. Size of the filled circles in each panel indicates the relative frequency of occurrence of each bin. Occurrence-weighted mean radiative susceptibility ($F_0$) is printed in red, under which is a decomposition of $F_0$ into precipitating-brightening (light green; positive susceptibility states with effective radii greater than 12 $\mu m$), entrainment-darkening (brown; negative susceptibility states and right-hand side of the EEF isoline), and Twomey-brightening (dark green; non-precipitating states with positive susceptibilities) regimes, with the occurrence of each regime in parentheses.





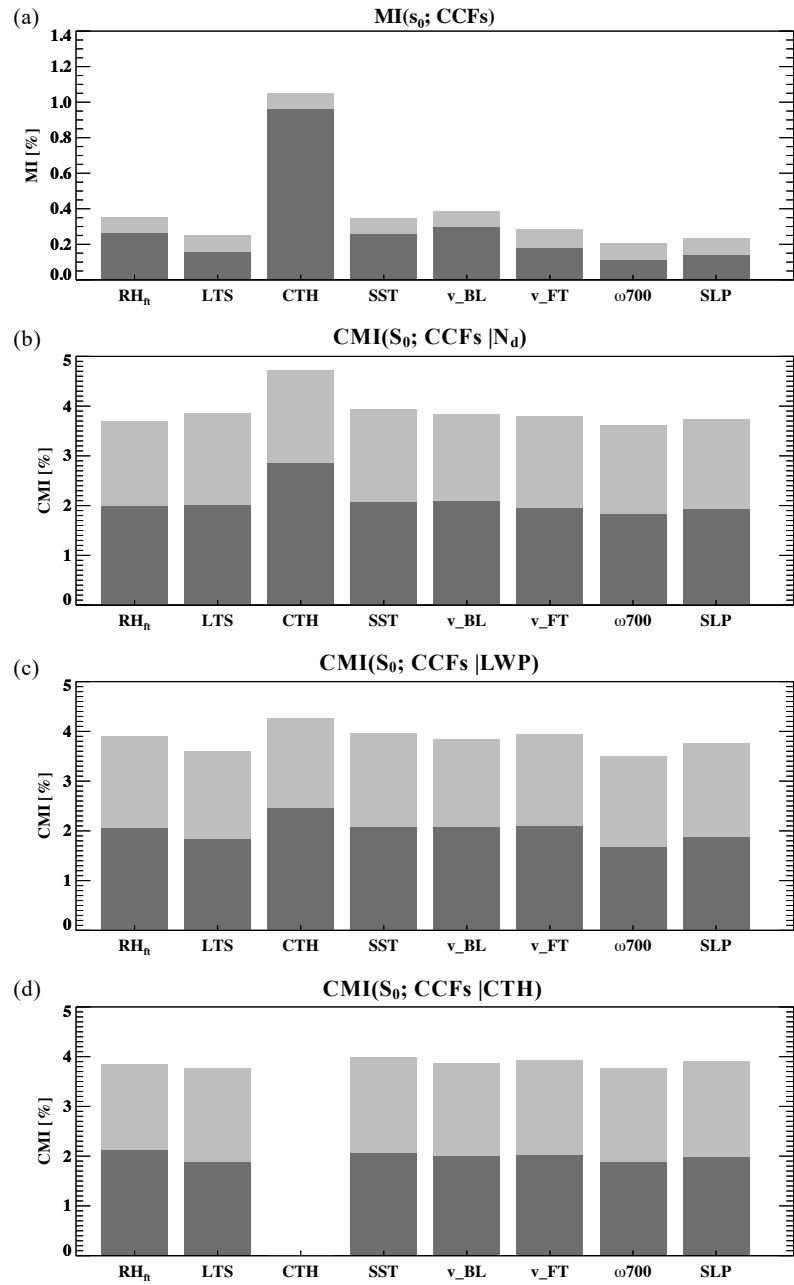

**Figure 3. a)** Mutual information (MI; dark gray) for $S_0$ and 8 meteorological factors (MFs), including $RH_{ft}$, LTS, CTH, SST, BL and FT winds, 700 hPa vertical velocity ($\omega 700$), and sea level pressure (SLP). **b)-d)** Conditional MI (CMI; dark gray) for $S_0$ and the 8 MFs, conditioned by $N_d$, LWP, and CTH, respectively. Noise-CMI (light gray) is represented by the (conditional) mutual information between MFs and randomly permuted $S_0$ sample space (effectively noise).



**Figure 4.** Mean meteorological/cloud state conditions associated with each LWP-$N_d$ bin in Fig. 2, in the space of **a)** CTH-SST, **b)** CTH-$RH_{ft}$, **c)** CTH-LTS, **d)** LTS-SST, **e)** LTS-$RH_{ft}$, and **f)** $RH_{ft}$-SST. Size and color of the circles represent the frequency of occurrence and the mean $S_0$ of that LWP-$N_d$ bin, respectively, as shown in Fig. 2. Precipitating clouds (based on a $r_e$ threshold of 12 $\mu m$) and non-precipitating clouds are indicated by filled and open circles, respectively.



**Figure 5.** Annual cycle of **a)** ERA5 $RH_{ft}$ (black), SST (blue), LTS (red), 700 hPa subsidence (gray), **b)** MODIS CTH (black), LWP (blue), and $N_d$ (red), as monthly means (filled circles), medians (filled triangles), and interquartile ranges (vertical bars). **c)** Annual cycle of occurrence-weighted $F_0$ (black) and the occurrence of each albedo susceptibility regime (colored).

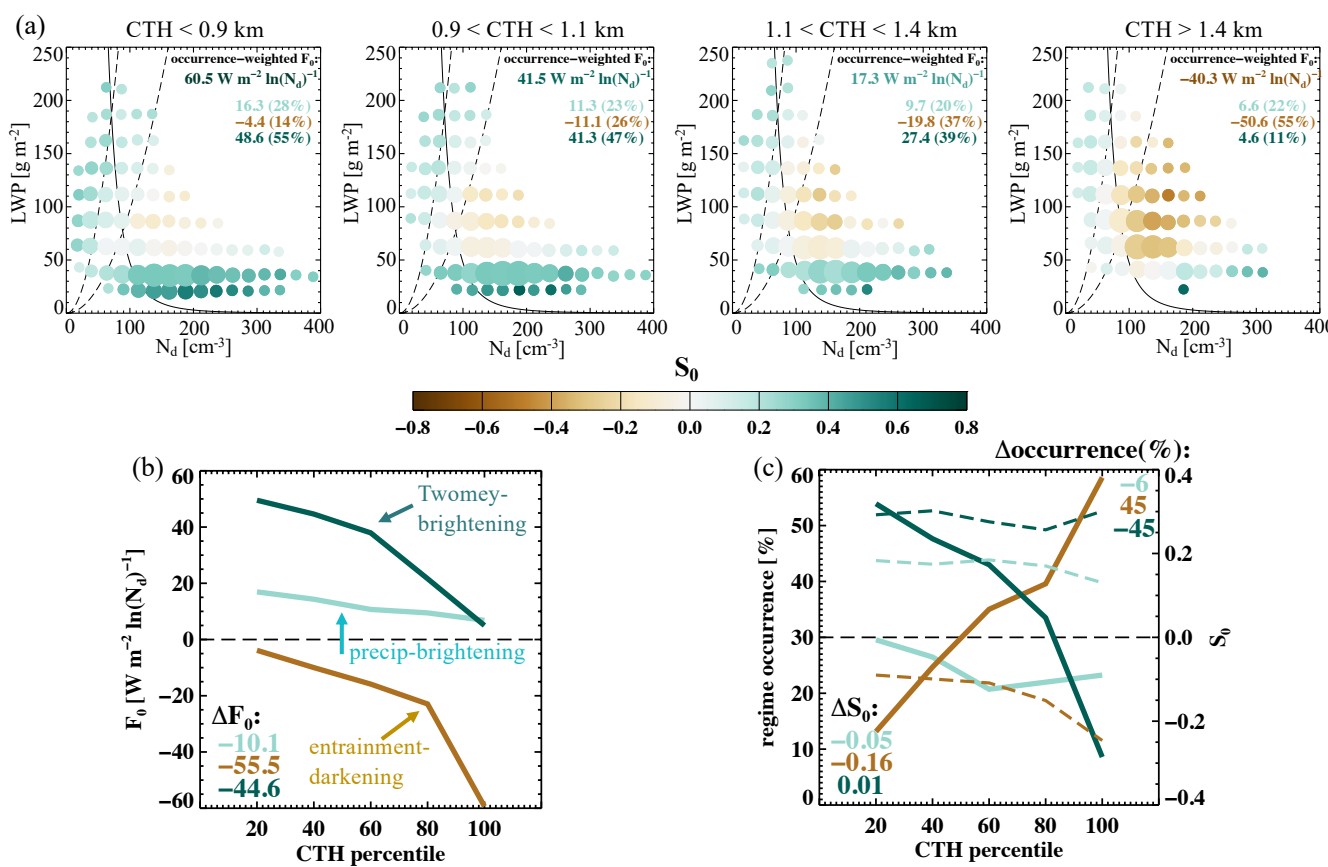

**Figure 6. a)** as in Fig. 2, but conditioned on cloud top height (CTH) quartiles. Occurrence-weighted mean **b)** $F_0$ and **c)** regime-mean $S_0$ (dashed curves) and regime-occurrence (solid curves) of the 3 albedo susceptibility regimes (defined in Fig. 2) as a function of CTH, increment of 20 percentile.





**Figure 7. a)-f)** as in Fig. 6, but conditioned on lower-tropospheric stability (LTS) and free-tropospheric relative humidity ($RH_{ft}$). Data are evenly divided into 6 equal-size LTS-$RH_{ft}$ bins.

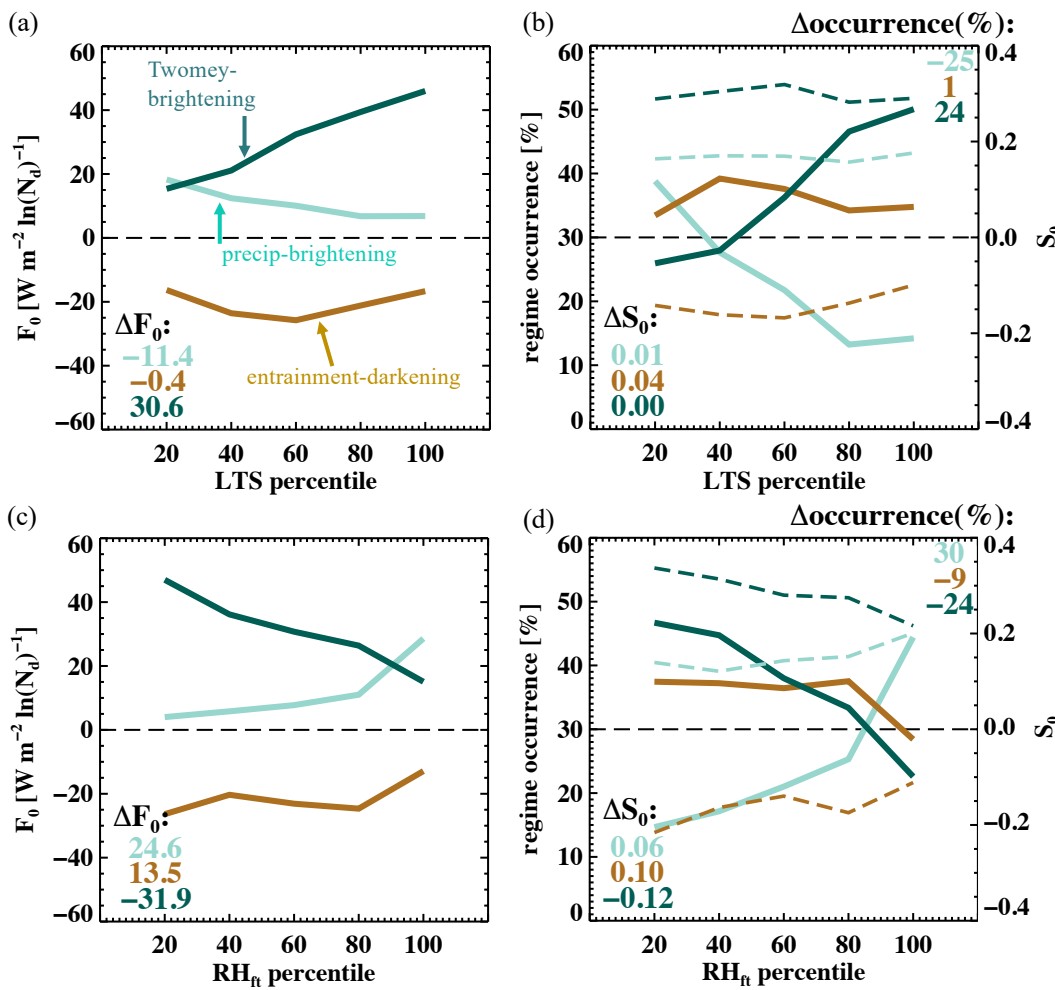

**Figure 8. a)-b)** and **c)-d)** as in Fig. 6b and c, but for LTS and $RH_{ft}$, respectively.

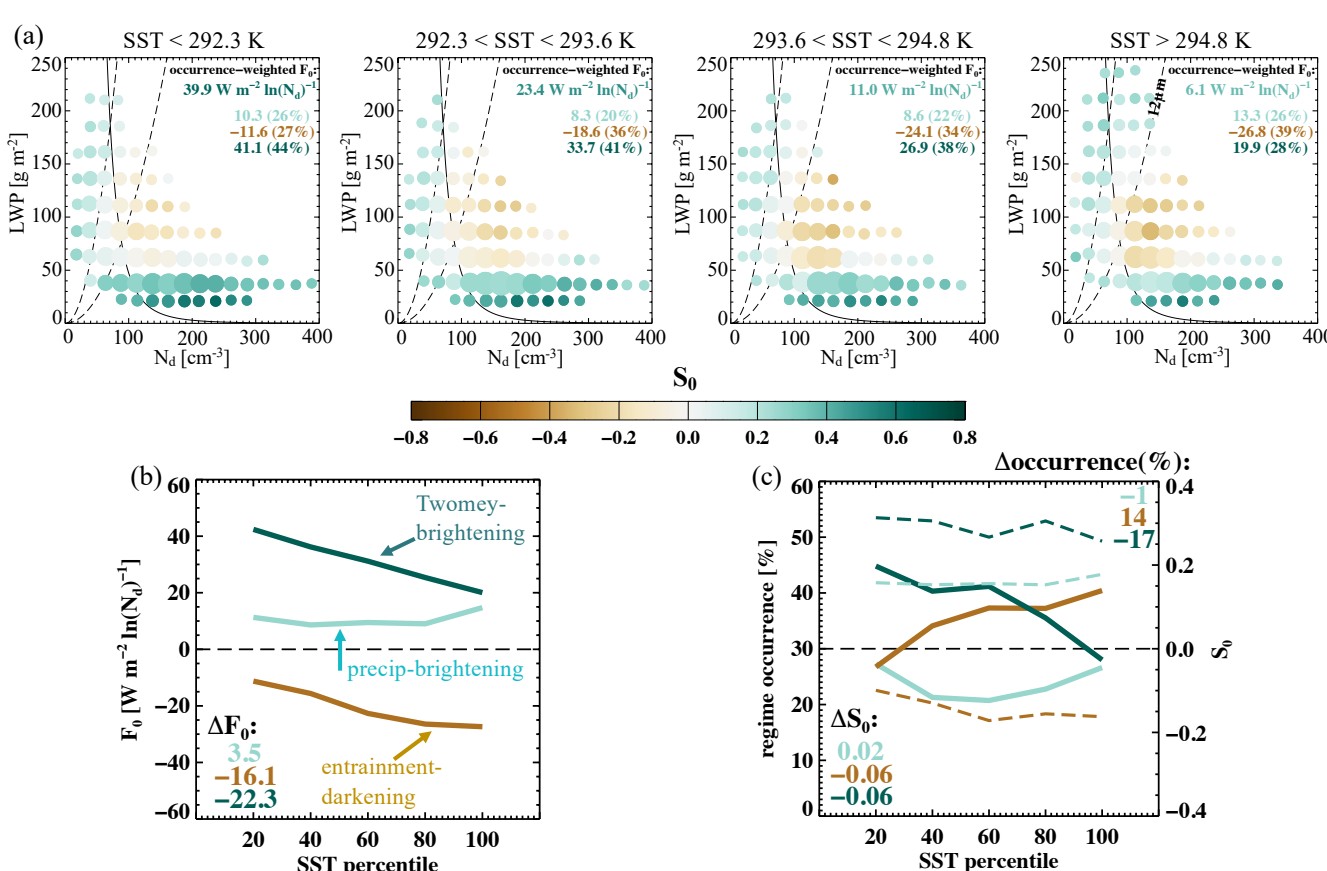

**Figure 9.** As in Fig. 6, but for sea surface temperature (SST).