# Peer review of "Albedo susceptibility of Northeastern Pacific stratocumulus: the role of covarying meteorological conditions"

_Atmospheric Chemistry and Physics, 2021_

## Referee Comment (RC1)

The authors aim to quantify an albedo and forcing susceptibility for stratocumulus in the NE Pacific by following Pearl 1984 and more recently Gryspeerdt's 2018 work by using Nd as a mediating variable in observed aerosol-cloud interactions. The authors find the albedo susceptibility exists in different regimes depending on the cloud state and environmental influence, with most occurring in the "Twomey brightening," "entrainment-darkening," and "precipitating-brightening."

While the study adds information on regime specific, aerosol-cloud interactions, it is difficult to follow at times. The study chooses an interesting region to focus on, where ACI are not always leading to a cooling effect, and these effects are heavily dependent on the environmental conditions. If the study can clarify its main message, and how the meteorological factors work to influence ACI on a process level, this would be a great resource for understanding aerosol-cloud-environmental interactions.

Major comments:

The authors explore the LWP and Nd space to begin, then rapidly transition into explaining the albedo and forcing susceptibilities. The authors say that using Nd mediates any confounding effects of the environment on observed patterns of aerosol-cloud interactions, however Nd would be similarly affected by environmental influences (though not to the same degree) as AOD.

When discussing each meteorological factor, the authors explain the observed patterns of the meteorological factor, while completely avoiding how these factors themselves initiate or influence the effects. There is too much emphasis on how each factor changes with the seasonal cycle, which obfuscates the message of the last section of the paper away from how these MF control ACI. The authors also rarely connect how the MF may work in tangent to influence ACI. Because you are working with real observations, the observed effects would have all MFs working in tandem on the cloud layer at all times.

The paper assumes CTH reflects the state of the cloud, and does not address the inherent relationship between increasing Nd and increasing CTH. CTH, and the deepening/invigoration effect aerosol has on shallow, warm clouds, would be an additional cloud adjustment effect. The authors should state or explore how increasing LWP, through precipitation suppression, may then act to deepen clouds.

Please cite studies to add more context to the forcing sensitivities given.

Minor comments:

Line 98: Is there any potential for sampling issues by using the Nd equation from Grosvenor et al. 2018 as opposed to other Nd or aerosol proxies?

Line 143: How does removing samples with low Ac and Nd correlations affect the validity of your results, since those scenes imply a null hypothesis?

Lines 173-175: How does weighting by CF affect the results? Does this introduce some type of bias that reduces the effects Nd has on smaller clouds like cumulus?

Line 192: Does this assume that higher Nd clouds have experienced some form of precipitation delay/suppression before precipitating?

Could using 1° x 1° means obscure less frequently occurring phenomena?

Line 243: What additional proof is there that entrainment-evaporation is occurring other than the daily means showing a slight "darkening?"

Line 269: If CTH reflects the cloud state and meteorological state, couldn't that be said of most cloud properties, as almost all cloud properties respond to LWP changes and MFs?

Line 323: If the precipitating-brightening regime is spread out along the meteorological space, could that indicate that you are missing a key meteorological factor that would explain this spread?

Section 4.3.3 How does LTS affect clouds and ACI on a process level?

Section 4.3.4a Does deepening occur due to aerosol, which then leads to increasing LWP, or does LWP increase first, which then deepens the cloud? Or does LTS decrease, increasing CTH, then LWP? Or do all of these occur simultaneously?

Section 4.34c How does stability work with the other MF?

Line 415: Please cite a reference for the comment on free tropospheric humidity conditions influencing LWP.

Section 4.3.4d How does SST influence the impact of other MF?

Line 459: Please add more discussion on confounding factors. Further, you state that because you are using Nd, you can infer a causal relationship, however this is not the case. Please clarify this in some way that using Nd alone does not constrain the effect of confounding factors.

---

## Referee Comment (RC2)

**Review of "Albedo susceptibility of Northeastern Pacific stratocumulus: the role of covarying meteorological conditions" by J. Zhang et al.**

Atmospheric Chemistry and Physics

October 18, 2021

Zhang et al. use satellite observations to study relationships between cloud droplet number concentration and cloud albedo in subtropical stratocumulus clouds. The observed relationships are interpreted in the context of several proposed mechanisms of cloud susceptibility to aerosol perturbations. Overall the topic and findings of the paper are important, and the paper is clearly written and fits within the scope of ACP. However, I have a few concerns about how the data are filtered and how uncertainty is quantified. For this reason, I recommend *major revision* for the manuscript. Please see my comments below.

**General Comments**

1. Line 105-100, 140. I understand the need to filter the data to get reliable retrievals of cloud-droplet number concentration (Nd), but I would have liked to see more discussion about potential sampling biases that could result from the filtering. First of all, what fraction of the data are you excluding by applying the filters? Do you believe that the inferred relationships approximately represent the entire cloud population in the study region or only a subset of the population? If the relationships only represent a subset of the population, then are the stated Nd-albedo relationships likely to underestimate or overestimate the true Nd-albedo relationship for the entire cloud population? Also, is it possible to have a cloud field that satisfies the filter criteria, then becomes exposed to an aerosol perturbation and no longer meets the filter criteria? One possible example is a cloud field that transitions between open and closed cellular convection regimes when exposed to an aerosol perturbation. Could such cases be an important component of the overall cloud-albedo susceptibility to aerosol perturbations? If so, then could the Nd-albedo relationships stated in the paper be biased, and in what direction?

2. Line 160. Do you account for spatial and temporal autocorrelation when estimating confidence intervals? If autocorrelation is not accounted for, then every observation is implicitly assumed to be an independent realization, which is not accurate. I would guess based on the published literature that neglecting autocorrelation in the uncertainty analysis would lead to confidence intervals that are a factor of 5-10 too narrow. I recommend estimating the spatial and temporal degrees of freedom following Bretherton et al. 1999 and then scaling the confidence intervals accordingly. For an example, see Myers et al. 2021.

3. Line 190. I believe that the dependence of cloud-droplet asymmetry parameter on droplet effective radius can make a non-negligible contribution to the total cloud-albedo dependence on effective radius when the droplets are relatively small (<12 um). I wonder if the assumption of constant asymmetry parameter could affect the Nd-albedo relationship in Fig. 1 for the high-Nd cases. Could you do a sensitivity test that defines asymmetry parameter as a function of mean droplet effective radius? The parameterization of Slingo 1989, or some approximation of it, is probably easier to implement than that of Hu and Stamnes 1993.

**Specific Comments**

- Line 5: "Aqua satellite, to …" remove comma
- Line 6: "low-cloud brightening potential of …" It would help to clarify that the stated value represents the albedo susceptibility for overcast scenes, not the area-average value.
- Line 41: "increase in LWP, that …" remove comma
- Line 125, 126: "casual" -> "causal"
- Line 177: "This means …" I suggest rephrasing this so that it does not use the word "means". This will avoid potential confusion with the alternate definition of "means" that relates to averages. This will be helpful since the word "average" is used later in the sentence.
- Line 230: "positive S0 reflects …" I suggest using a word other than "reflects" here to avoid potential confusion with the alternate meaning of "reflects" that is associated with radiation.
- Line 371: Please report uncertainties. This is very important information because it facilitates comparison with other work, among other things.

**References**

Bretherton, C. S., Widmann, M., Dymnikov, V. P., Wallace, J. M., & Bladé, I. (1999). The Effective Number of Spatial Degrees of Freedom of a Time-Varying Field, *Journal of Climate*, *12*(7), 1990-2009. Retrieved from https://journals.ametsoc.org/view/journals/clim/12/7/1520-0442_1999_012_1990_tenosd_2.0.co_2.xml

Hu, Y. X., & Stamnes, K. (1993). An Accurate Parameterization of the Radiative Properties of Water Clouds Suitable for Use in Climate Models, *Journal of Climate*, *6*(4), 728-742. Retrieved from https://journals.ametsoc.org/view/journals/clim/6/4/1520-0442_1993_006_0728_aapotr_2_0_co_2.xml

Myers, T.A., Scott, R.C., Zelinka, M.D. *et al.* Observational constraints on low cloud feedback reduce uncertainty of climate sensitivity. *Nat. Clim. Chang.* **11,** 501–507 (2021). https://doi.org/10.1038/s41558-021-01039-0

Slingo, A. (1989). A GCM Parameterization for the Shortwave Radiative Properties of Water Clouds, *Journal of Atmospheric Sciences*, *46*(10), 1419-1427. Retrieved from https://journals.ametsoc.org/view/journals/atsc/46/10/1520-0469_1989_046_1419_agpfts_2_0_co_2.xml

---

## Author Comment (AC1)

**Response to reviews of "Albedo susceptibility of Northeastern Pacific stratocumulus: the role of covarying meteorological conditions" by J. Zhang et al.**

We would like to thank Alyson Douglas and two other anonymous reviewers for their constructive comments and suggestions on our manuscript, which encouraged us to think more deeply about our analysis and helped us improve the original manuscript.

Specific responses to each comment are contained below, with the reviewers' comments provided in blue and our responses in black. Changes to the manuscript made in response to the reviewer are provided in red italics. We have also made unsolicited smaller changes to the manuscript to further polish the writing.

**REVIEWER 1 (Alyson Douglas):**

The authors aim to quantify an albedo and forcing susceptibility for stratocumulus in the NE Pacific by following Pearl 1984 and more recently Gryspeerdt's 2018 work by using Nd as a mediating variable in observed aerosol-cloud interactions. The authors find the albedo susceptibility exists in different regimes depending on the cloud state and environmental influence, with most occurring in the "Twomey brightening," "entrainment-darkening," and "precipitating-brightening."
While the study adds information on regime specific, aerosol-cloud interactions, it is difficult to follow at times. The study chooses an interesting region to focus on, where ACI are not always leading to a cooling effect, and these effects are heavily dependent on the environmental conditions. If the study can clarify its main message, and how the meteorological factors work to influence ACI on a process level, this would be a great resource for understanding aerosol-cloud-environmental interactions.

We thank Dr. Douglas for these comments and helpful suggestions. We revised the introduction in order to convey the key message and the goal of this study upfront, to help better guide the readers as they proceed through the rest of the paper. Both in the revised Introduction and Methods sections, we now make it clear that this study differs from those that focus on individual controlling factors and the evolution of a cloud system by attempting to untangle aerosol and meteorological effects (**please see more details in the revised sections and specific responses below**). The top-down approach we've taken allows us to understand and quantify the impact of meteorological conditions as a *collective driver* of cloud albedo susceptibility, taking into account the covariabilities among them.

The manuscript has been revised to clarify the issues and concerns raised accordingly. Direct responses to comments and changes to the original manuscript are listed below.

Major comments:

The authors explore the LWP and Nd space to begin, then rapidly transition into explaining the albedo and forcing susceptibilities. The authors say that using Nd mediates any confounding effects of the environment on observed patterns of aerosol-cloud interactions, however Nd would be similarly affected by environmental influences (though not to the same degree) as AOD.

Exploring cloud albedo susceptibility in the LWP and Nd space is the key feature of this study, and we present this by first showing the climatology of Nd-LWP-Ac relationship in this space. The benefits of using this framework have recently been demonstrated by Glassmeier et al. (2019, 2021) and Hoffmann et al. (2020). The latter paper in particular showed how one can link features in Nd-LWP space to physical processes. Hence it connects between the top-down and bottom-up (process) views of the cloud system.

Regarding using Nd as an intermediary to reduce meteorological confounding effects, it has been shown by Gryspeerdt et al. (2016,2019) to be effective when exploring the causal relationship between aerosol and cloud properties. We agree with the reviewer that environmental influences could still be at play even when we use Nd, and the way we further constrain meteorological confounding effects is by using satellite snapshots where we argue that meteorological conditions are approximately homogenous within the limited space-time frame (a satellite snapshot at 1° by 1°). This is not saying that meteorology does not play a role in our study, but rather that when we calculate/regress albedo susceptivity on a 1° x 1° grid, meteorology is controlled. Furthermore, key to this study, as expressed in the title, the goal is to understand and quantify the influence of covarying meteorological conditions on albedo susceptibility.

**The Methods section has been largely reconstructed to clarify this methodology. This includes but is not limited to the text added below:**

*"In the case of satellite observations, confounding factors can be significantly reduced: for a given satellite snapshot (e.g. covering a 1◦ × 1◦ area), meteorological conditions can be assumed homogeneous within a limited space-time frame, enabling one to relate changes in cloud radiative properties to respective changes in Nd (e.g. Goren and Rosenfeld, 2014; Painemal, 2018)."*

*"We begin our results section by introducing an informative parameter space, the LWP-Nd space. The choice of these variables is motivated by mutual information analyses that help establish the dominating role of LWP and Nd in governing albedo susceptibility. Exploring the behavior of these high-level fingerprints (LWP-Nd) of the system is a pathway to bridge and balance between the Newtonian and Darwinian approaches that will benefit our understanding on the multi-scale and multidisciplinary nature of the aerosol-cloud system (Mülmenstädt and Feingold, 2018)."*

When discussing each meteorological factor, the authors explain the observed patterns of the meteorological factor, while completely avoiding how these factors themselves initiate or

influence the effects. There is too much emphasis on how each factor changes with the seasonal cycle, which obfuscates the message of the last section of the paper away from how these MF control ACI. The authors also rarely connect how the MF may work in tangent to influence ACI. Because you are working with real observations, the observed effects would have all MFs working in tandem on the cloud layer at all times.

This comment is very related to the main message of this study, and we appreciate the suggestion to make it clear in the introduction. **We have reconstructed the last few paragraphs of the introduction to convey the main focus/message/goal of this study upfront, to help better guide readers through the rest of the paper.**

First of all, we did not position our study as an addition to the already rich pool of existing process-oriented studies on how individual meteorological factors affect aerosol-cloud interactions. Because of the nature of these process studies, one has to constrain the variability of the system in terms of one meteorological factor while minimizing variability in the others. No doubt these are valuable and carefully designed studies that have made great contributions by helping the community to understand the role of individual ACI controlling factors. That said, our study has taken a different perspective where we embrace the covariabilities between these factors (the missing piece of information in these previous process studies) for two main reasons: i), as the reviewer points out "Because you are working with real observations, the observed effects would have all MFs working in tandem on the cloud layer at all times," nature always presents a combination of environmental conditions that work together to affect ACI and we have to understand the collective impact of this covariability and quantify it in various ways (Feingold et al. 2016; Mülmenstädt and Feingold 2018); ii) with respect to radiative forcing/effect, one has to take into account the frequency with which these meteorological conditions occur, which requires us to quantify the covariability between these meteorological effects.

**We have also re-organized the structure of our Results section to be coherent with conveying this main message.**

The paper assumes CTH reflects the state of the cloud, and does not address the inherent relationship between increasing Nd and increasing CTH. CTH, and the deepening/invigoration effect aerosol has on shallow, warm clouds, would be an additional cloud adjustment effect. The authors should state or explore how increasing LWP, through precipitation suppression, may then act to deepen clouds.

We appreciate this comment. First of all, when we discuss state of the cloud or cloud states, e.g. LWP, Nd, CTH, we did not imply they represent independent aspects of the clouds or they can independently represent a cloud state; they are all well correlated with one other.

Second, we want to clarify that the methodology used in this study allows us to explore/infer ACI as presented by satellite snapshots of cloudy scenes, meaning whatever we observe represents the current state of the cloud system (a Markovian methodology). However, as we stated in the **revised Introduction and Method sections**, we have to refrain from interpreting our results in

terms of the evolution of the system, as the current methodology doesn't allow us to do so. Therefore, the deepening aspect of the precip-suppression process is not a target of this study and more suited for an evolution-oriented study, e.g. LES simulations or utilizing geostationary satellite observations.

Lastly, we want to clarify that we mainly use CTH as a proxy for indicating *a meteorological condition*, that is the depth of the marine boundary layer, i.e., the highest level that cloud tops can reach. Wording to this point has been revised and clarified in the revised manuscript accordingly.

*"Cloud top heights of marine stratocumulus, marine boundary layer heights, and inversion heights are positively correlated in the setting of the stratocumulus-topped boundary layer (STBL) over the NE Pacific. Therefore, CTH is considered here as a variable indicating one aspect of the cloud states, similar to LWP, while concurrently serving as an indication of a meteorological condition, namely the depth of the STBL."*

Please cite studies to add more context to the forcing sensitivities given.

We cited Douglas and L'Ecuyer (2019) and Painemal (2018) who report this type of forcing per Nd perturbation metric in their studies, although we note that caution should be applied when making direct comparisons due to slight differences in data selection and methodology. We choose not to provide any further references/studies that report radiative forcings/effects in W m$^{-2}$, as it would not make a fair comparison and may lead to further confusion.

Minor comments:

Line 98: Is there any potential for sampling issues by using the Nd equation from Grosvenor et al. 2018 as opposed to other Nd or aerosol proxies?

We have checked the consistency of our Nd-retrieving method with Grosvenor and Wood (2014,2018), Gryspeerdt et al. (2016,2019), and Painemal (2018), and there are no potential sampling issues that we are aware of. Our choice to focus on unbroken clouds at the 20 km scale means that we avoid the high Reff/low cloud optical depth biases associated with MODIS retrieval of broken cloud properties. No doubt uncertainties due to assumptions about adiabatic fraction and distribution width exist, but this is the nature of satellite-based Nd retrievals.

Line 143: How does removing samples with low Ac and Nd correlations affect the validity of your results, since those scenes imply a null hypothesis?

As the reviewer points out, low Ac-Nd correlations suggest calculations with rather low level of statistical significance, and the threshold of 0.2 we chose is a trade-off between increasing statistical significance and keeping a reasonable number of scenes to analyze. Figure 1 below shows the result without the $|r(Ac,Nd)|>0.2$ filtering, and one can see that this does not change the qualitative aspect of our results, but does change the overall occurrence-weighted $F_0$ a little from 17.0 (no $|r|$ threshold) to 20.8 ($|r|>0.2$) W m$^{-2}$ ln(Nd)$^{-1}$.

We have revised the wording around this, which now reads as: *"$S_0$ values are only reported if the number of data points is greater than or equal to 5 and the absolute value of the correlation coefficient if greater than 0.2. This provides levels ranging from 25% (minimum required number of samples, 5) to 60% (maximum number of samples within a 1° grid) at which the correlations are statistically significantly according to a Student's t-test. Applying such a threshold on the absolute value of the correlation coefficient between Ac and Nd shrinks the sample size of $S_0$ by ~19% but does increase the statistical significance of the results by at least 25%. A sensitivity test using $S_0$ without the correlation coefficient threshold (not shown) indicates no qualitative impacts on the results but a subtle quantitative impact on the occurrence-weighted $F_0$ (introduced below; from 20.8 to 17.0 W m-2 ln(Nd)-1)."*

[Figure]

*Figure 1. Same as in Fig. 2 of the manuscript, but for all correlation coefficients (without |r| threshold).*

Lines 173-175: How does weighting by CF affect the results? Does this introduce some type of bias that reduces the effects Nd has on smaller clouds like cumulus?

We appreciate this question as we realized the original discussion on this 1-degree averaging is a bit confusing. We reconstructed this paragraph to clarify it, which now reads as *"… we aggregate cloud properties, including cloud albedo, and ERA5 meteorological variables (during the Aqua overpass over a 2-hour period) to the same 1° x 1° grid on which $S_0$ is calculated. The aggregation method follows a straightforward arithmetic mean of all the pixel-level data points within the grid (0.25° for ERA5 and 20-km for MODIS-CERES), except for cloud properties where we only select overcast footprints for averaging, because Nd is only retrieved at overcast footprints."*

Basically, because Nd is only retrieved under overcast conditions, the CF weighting doesn't have any effect, and we have removed the CF weighting phrase in the revised sentence to avoid confusion.

Line 192: Does this assume that higher Nd clouds have experienced some form of precipitation delay/suppression before precipitating?

No, we don't think one can deduce this directly from this LWP-Nd space, which is built on satellite snapshots of cloudy scenes. One has to turn to an evolution-oriented study (e.g. LES or geostationary satellite based) to prove this assumption.

Could using 1° x 1° means obscure less frequently occurring phenomena?

The 1° x 1° grid is used to constrain for sub-grid meteorological variability; besides, our region of study is only a 10° x 10° area, which limits larger areal averages. If the reviewer meant "low-frequency variability" by "less frequently occurring phenomena," we have shown how susceptibility varies at a seasonal (month-to-month) temporal scale.

Line 243: What additional proof is there that entrainment-evaporation is occurring other than the daily means showing a slight "darkening?"

First of all, each circle here, e. g. a brownish circle, represents a LWP-Nd bin-mean susceptibility using 10-years of satellite observations, i.e., it does not represent a daily mean.

In response to the reviewer's comment on additional proof:

i)      we plotted an evaporation-entrainment feedback (EEF) isoline on the figure, which indicates that to the right of the line, droplets take less time (< 3s) to evaporate, and the fact that the brownish circles align extremely well with this isoline, is considered strong support of the entrainment-evaporation thinking. One can compare this to Hoffmann et al. (2020) who emulated dLWP/dt in Nd-LWP space based on a very large ensemble of LES simulations;

ii)     ii) we further plotted dln(LWP)/dln(Nd) on the same LWP-Nd space (Fig. 2 below), and one can see clear negative LWP responses to Nd perturbations in this part of the LWP-Nd space, as further substantiation of this result.

[Figure]

*Figure 2. Same as in Fig. 2 of the manuscript, but for dln(LWP)/dln(Nd).*

Line 269: If CTH reflects the cloud state and meteorological state, couldn't that be said of most cloud properties, as almost all cloud properties respond to LWP changes and MFs?

Yes, it is true that almost all cloud properties respond to changes in MFs, however, this is not what we meant here; we meant that the CTH used in our analyses is considered as an indication of cloud states, as it correlates very well with cloud LWP, while also serving as a proxy for meteorological conditions – in this case the depth of the marine boundary layer.
We have reworded this sentence to clarify the point in the revised manuscript:
*"Cloud top heights of marine stratocumulus, marine boundary layer heights, and inversion heights are positively correlated in the setting of the stratocumulus-topped boundary layer (STBL) over the NE Pacific. Therefore, CTH is considered here as a variable indicating one aspect of the cloud states, similar to LWP, while concurrently serving as an indication of a meteorological condition, namely the depth of the STBL"*

Line 323: If the precipitating-brightening regime is spread out along the meteorological space, could that indicate that you are missing a key meteorological factor that would explain this spread?

First, we do not aim to find one-to-one matchings between $S_0$ regimes and a MET variable that explains it or controls where the regime should fall in the MET spaces. In fact, we don't believe there is always one MET factor that explains all. This echoes our main message of this paper, that is the importance of understanding the covariability between MET conditions and the role it plays in affecting the cloud albedo susceptibility.

Second, the fact that two $S_0$ regimes cluster while the other one spreads out serves to expand our discussion on the concept of "equifinality", i.e. different initial/boundary conditions can yield the same realization, in our case, meaning different MET conditions can produce the same $S_0$.

We added a sentence to emphasize this point in the revised manuscript:
*"The fact that two of the susceptibility regimes cluster while the other one spreads out in the meteorological spaces serves to expand our discussion on the concept of "equifinality". We previously discussed that different meteorological conditions may produce the same cloud state (LWP, Nd). Here we see that different meteorological conditions may produce the same $S_0$. This ties back to the importance of understanding and quantifying the covariabilities between meteorological factors, as multiple environmental factors may be needed to explain all the variability in cloud states (e.g. Chen et al. 2021) and thereby albedo susceptibility."*

Section 4.3.3 How does LTS affect clouds and ACI on a process level?

As we now clarify our main goal/message upfront in the Introduction section of the revised manuscript, thanks to the comments from the reviewer, we believe we have made clear that our top-down approach focuses on the collective role of MFs (covarying MET conditions) on albedo susceptibility, instead of trying to untangle individual meteorological effect, e.g. "How does LTS affect clouds and ACI on a process level?"

But to directly answer the reviewer's question on LTS, as has been shown by many previous studies (e.g. Chen et al. (2014), Gryspeerdt et al. (2019)), the role of LTS in ACI is mainly through affecting the cloud top entrainment and the cloud top radiative cooling, such that strong LTS leads to less entrainment and stronger cloud-top cooling. One has to also keep in mind that for eastern subtropical stratocumulus regions, when LTS is strong, SST is usually cool, and the free-troposphere is typically dry with enhanced subsidence — another clear demonstration of considering covariability.

We touched upon on this aspect in the original Section 4.3.4 (now Section 5.3 b. & c.), but rather focus on how the cloud states shift in LWP-Nd space going from high to low LTS (with RHft covarying with LTS). We refer the reviewer to our revised Section 5.3 (b) for discussion on the role of LTS on albedo susceptibility and how does the negatively correlated RHft conditions modulate the influence of LTS.

Section 4.3.4a Does deepening occur due to aerosol, which then leads to increasing LWP, or does LWP increase first, which then deepens the cloud? Or does LTS decrease, increasing CTH, then LWP? Or do all of these occur simultaneously?

We want to clarify that the methodology used in this study allows us to explore/infer ACI as presented by satellite snapshots of cloudy scenes, meaning whatever we observe represents the current state of the cloud system (a Markovian methodology). However, as we stated in the revised Introduction and Method sections, we have to refrain from interpreting our results to inform the temporal evolution of the system, as the current methodology doesn't allow us to do

so. Therefore, this type of question is more suited to an evolution-oriented study, e.g. LES simulations or utilizing geostationary satellite observations.

The CTH conditioning here merely serve as a proxy for separating cloudy scenes that reside in shallow to deep marine boundary layers.

**Section 4.34c How does stability work with the other MF?**

For eastern subtropical stratocumulus regions, when LTS is strong, SST is usually cool, and the free-troposphere is typically dry with enhanced subsidence. Figure 4 (manuscript) shows how these variables tend to covary in 2-D slices through the data while Figure 5 shows the seasonal co-evolution of these variables and their impact on $F_0$.

**Line 415: Please cite a reference for the comment on free tropospheric humidity conditions influencing LWP.**

References to Ackerman et al. (2004) Science and Chen et al. (2014) Nature Geo are added.

**Section 4.3.4d How does SST influence the impact of other MF?**

As a result of the seasonal evolution of the large-scale circulation pattern over the northeast Pacific, when SST is warm, LTS is typically low and RHft is usually high. Figure 4 (manuscript) shows how these variables tend to covary in 2-D slices through the data while Figure 5 (manuscript) shows the seasonal co-evolution of these variables and their impact on $F_0$.

**Line 459: Please add more discussion on confounding factors. Further, you state that because you are using Nd, you can infer a causal relationship, however this is not the case. Please clarify this in some way that using Nd alone does not constrain the effect of confounding factors.**

We appreciate this comment which made us revisit our Methods section and further clarify our approaches, reasoning behind choices, and assumptions applied (including the causal relationship, the choice of using Nd, and how confounding factors are further constrained in our analyses).

The use of Nd as an intermediate variable is to reduce the confounding effects of meteorology on the causal relationship between aerosol and cloud properties, similar to Gryspeerdt et al. (2016, 2019). We now clarify that in the case of our satellite observation-based study, *"confounding factors can be significantly reduced: for a given satellite snapshot (e.g. covering a 1◦ × 1◦ area), meteorological conditions can be assumed homogenous within a limited space-time frame, enabling one to relate changes in cloud radiative properties to respective changes in Nd (e.g. Goren and Rosenfeld, 2014; Painemal, 2018)."*
**Please see the revised Methods section for detailed clarifications.**

**REVIEWER 2:**

Zhang et al. use satellite observations to study relationships between cloud droplet number concentration and cloud albedo in subtropical stratocumulus clouds. The observed relationships are interpreted in the context of several proposed mechanisms of cloud susceptibility to aerosol perturbations. Overall the topic and findings of the paper are important, and the paper is clearly written and fits within the scope of ACP. However, I have a few concerns about how the data are filtered and how uncertainty is quantified. For this reason, I recommend *major revision* for the manuscript. Please see my comments below.

We thank the reviewer for the encouraging words and helpful suggestions/comments. The manuscript has been revised accordingly to clarify the concerns on the uncertainty quantification and data filtering. Direct responses to comments and changes to the original manuscript are listed below.

General Comments

1. Line105-100, 140. I understand the need to filter the data to get reliable retrievals of cloud-droplet number concentration (Nd), but I would have liked to see more discussion about potential sampling biases that could result from the filtering. First of all, what fraction of the data are you excluding by applying the filters?

Among all the 10-year Aqua-sample 1-degree scenes, ~53% of them consist of single-layer liquid clouds ONLY. Among these single-layer liquid cloud scenes, ~41% of them have a reported $S_0$ value (meaning they satisfy the Nd and $S_0$ calculation criteria).
We added a sentence to include this information in the revised manuscript: *"For 1°x1° satellite sampled scenes, ~53% consist of single-layer liquid clouds only, and among these cloudy scenes, ~41% satisfy the Nd and $S_0$ calculation criteria (introduced in Section 3) and are subsequently used in this study."*

Do you believe that the inferred relationships approximately represent the entire cloud population in the study region or only a subset of the population? If the relationships only represent a subset of the population, then are the stated Nd-albedo relationships likely to underestimate or overestimate the true Nd-albedo relationship for the entire cloud population?

The inferred relationships in our study represent high cloud fraction (> 0.8 at 1° scale) scenes of the Northeastern Pacific stratocumulus clouds, whose contribution to the overall cloud radiative effect of the entire cloud population of this region is significant compared to the rest of the population (less cloudy).

First, in this study, we focus on high cloud fraction (fc) marine stratocumulus clouds, because of the limitations to Nd-retrieving capability. Moreover, because the $1\circ \times 1\circ$ cloud fractions of these cloudy scenes analyzed in this work are high, their contribution to the overall cloud radiative effect of the entire cloud population of this region is significant compared to the rest of the

population (less cloudy). Thus, it is important and informative to quantify the response of these high-fc clouds to aerosol perturbations.

But more importantly, it is not the goal of this study to generalize our results to all marine stratocumulus clouds, especially those with thin optical depth, broken/open-cellular structures, with high sub-pixel inhomogeneity.

Languages reflecting these points and thinking are added to the revised manuscript: *"Because the 1∘ × 1∘ cloud fractions of these cloudy scenes analyzed in this work are high (comprising ~41% of all single-layer liquid cloud scenes over this region), their contribution to the overall cloud radiative effect of the entire cloud population of this region is significant, compared to the rest of the population (less cloudy). Thus, it is important and informative to quantify the response of these high-fc clouds to aerosol perturbations. That said, it is not the goal of this study to generalize the albedo susceptibility assessment presented here to all marine stratocumulus clouds, especially those with low optical depth, broken or open-cellular structure (high sub-pixel inhomogeneity), conditions under which space-borne Nd retrievals are highly uncertain (Grosvenor et al., 2018)."*

Also, is it possible to have a cloud field that satisfies the filter criteria, then becomes exposed to an aerosol perturbation and no longer meets the filter criteria? One possible example is a cloud field that transitions between open and closed cellular convection regimes when exposed to an aerosol perturbation. Could such cases be an important component of the overall cloud-albedo susceptibility to aerosol perturbations? If so, then could the Nd-albedo relationships stated in the paper be biased, and in what direction?

We thank the reviewer for this comment. First, we want to clarify that the methodology used in this study allows us to explore/infer ACI as presented by satellite snapshots of cloudy scenes, meaning whatever we observe represents the current state of the cloud system (a Markovian methodology). Therefore, we have to refrain from interpreting our results to inform the evolution of the system, e. g. transitions between open and closed cellular convection regimes, as the current methodology doesn't allow us to do so.

That said, we do acknowledge that cloud fraction changes to Nd perturbations contributes to the **scene** albedo susceptibility, but that is not a target of the current study, which focuses on **cloud** albedo susceptibility. Moreover, a recent study (Watson-Parris et al. 2021) has shown that the effective radiative impact of closing all pockets of open cells globally may be as small as ~0.01 W m$^{-2}$.

**These responses to comment 1 are reflected in the revised manuscript as part of the re-writing of the Introduction and Methods sections.**

2. Line160. Do you account for spatial and temporal autocorrelation when estimating confidence intervals? If autocorrelation is not accounted for, then every observation is implicitly assumed to be an independent realization, which is not accurate. I would guess based on the published literature that neglecting autocorrelation in the uncertainty analysis would lead to confidence

intervals that are a factor of 5-10 too narrow. I recommend estimating the spatial and temporal degrees of freedom following Bretherton et al. 1999 and then scaling the confidence intervals accordingly. For an example, see Myers et al. 2021.

We thank the reviewer for raising this point regarding the spatial-temporal autocorrelation when estimating regression uncertainties. We use the methods of Bretherton et al. (1999) to calculate the effective degree of freedom of $A_c$ within a 1 x 1 degree box and conclude that on average ~1.3 out of 10 footprints is independent. As a result, we scale up our uncertainties by ~2.8. The revised text around the estimating confidence intervals now reads as *"… We then further scale the uncertainty associated with the regression slopes by the square root of the ratio of the nominal to effective degree of freedom of Ac within 1° × 1° grid boxes to account for the spatiotemporal autocorrelation associated with the regressed field, similar to Myers et al. (2021). We compute the average value of effective degree of freedom using 10 years of CERES data covering the 10° × 10° study area and the methods of Bretherton et al. (1999). …"*

3. Line190. I believe that the dependence of cloud-droplet asymmetry parameter on droplet effective radius can make a non-negligible contribution to the total cloud-albedo dependence on effective radius when the droplets are relatively small (<12 um). I wonder if the assumption of constant asymmetry parameter could affect the Nd-albedo relationship in Fig. 1 for the high-Nd cases. Could you do a sensitivity test that defines asymmetry parameter as a function of mean droplet effective radius? The parameterization of Slingo 1989, or some approximation of it, is probably easier to implement than that of Hu and Stamnes 1993.

We thank the reviewer for pointing out the dependence of cloud-droplet asymmetry parameter on droplet effective radius.

We expect the impact of including $r_e$-dependence on the g parameter will have a small impact overall. Actually, we expect the impact to be more noticeable for low-Nd (bigger droplet) cases. We also want to clarify that the purpose of using the two-stream approximation is only for SZA adjustment on the measured cloud albedo (from measured SZA to 0°), meaning we use the two-stream approximation to get tau-dependent Ac-SZA relationships, **and we did not use it to directly calculate cloud albedo at all**.

We did a sensitivity test as suggested, using the linear parameterization of Slingo 1989 for the visible wavelength, and obtained the results as shown in Figure 3 below. As expected, including $r_e$-dependence on the g parameter only noticeably affects a few albedo points in the dark gray area ($r_e$ > 15 micron) to some extent whereas differences for smaller drop sizes are almost negligible, and the Nd-albedo relationship remain almost the same.

As a result of this sensitivity test, we decided to use the suggested $r_e$-dependence g parameter, as it is a more accurate representation of the two-stream approximation.

[Figure]

*Figure 3. Same as in Fig. 1 of the manuscript, but overlaid with blue open circles indicating an effective radius dependent scattering asymmetry parameter, adopted from Slingo (1989), for SZA adjustment.*

The wording regarding this SZA adjustment has been revised and now reads as *"Moreover, the cloud albedos used in this particular analysis are adjusted to an overhead solar zenith angle (SZA = 0°), in order to obtain a consistent basis for Ac-LWP-Nd relationships. This is done using the two-stream approximation (Meador and Weaver, 1980), which relates cloud albedo to cloud optical depth and solar zenith angle. Therefore, for a given τ we can obtain a theoretical Ac-SZA relationship using the two-stream approximation. The scattering asymmetry parameter is approximated by a linear function of re following Slingo (1989). We then use the theoretical τ-dependent Ac-SZA relationships to adjust Ac from measured SZA to overhead SZA."*

Specific Comments

• Line 5: "Aqua satellite, to …" remove comma

Done.

• Line 6: "low-cloud brightening potential of …" It would help to clarify that the stated value represents the albedo susceptibility for overcast scenes, not the area-average value.

We want to clarify that our strict fc > 0.99 criteria is applied at the footprint (20-km) level, not at the 1-degree level (where we calculate albedo susceptibility), therefore, even though we only include overcast footprints, 1-degree cloud fraction can still be less than 1, meaning partly cloudy

(1-degree) cases are included in our analyses, but restricted to high cloud fraction at the 1-degree scale. In fact, Figure 4 below shows the distribution of cloud fraction of 1-degree scenes used in our analyses, and more than 96% of the 1-degree scenes used in our study have a cloud fraction greater than 0.8 (or 99% of fc > 0.6), with only ~35% are overcast scenes at the 1 degree x 1 degree scale.

We included this Figure 4 as Fig. S1 in the supplementary material of the revised manuscript. We also added sentences at the end of the Method Section to clarify this concern *"Note that requiring overcast conditions for Nd retrievals at footprint level does not restrict the 1° × 1° cloudy scenes analyzed in this study to only overcast scenes, meaning partly cloudy scenes are included in our analyses. In fact, only ~35% of our 1° cloudy scenes are overcast (see the distribution of 1° × 1° cloud fraction in Fig. S1). Because the 1° × 1° cloud fractions of these cloudy scenes analyzed in this work are high (comprising ~41% of all single-layer liquid cloud scenes over this region), their contribution to the overall cloud radiative effect of the entire cloud population of this region is significant, compared to the rest of the population (less cloudy). Thus, it is important and informative to quantify the response of these high-fc clouds to aerosol perturbations. That said, it is not the goal of this study to generalize the albedo susceptibility assessment presented here to all marine stratocumulus clouds, especially those with low optical depth, broken or open-cellular structure (high sub-pixel inhomogeneity), conditions under which space-borne Nd retrievals are highly uncertain (Grosvenor et al., 2018)."*

[Figure]

*Figure 4. Probability distribution of 1-degree low cloud fraction (LCF) from the cloudy scenes that satisfy the criteria for albedo susceptibility regression. Note the log scale on y-axis.*

- Line 41: "increase in LWP, that ..." remove comma

Done.

• Line 125, 126: "casual" -> "causal"

Done. Thanks!

• Line 177: "This means ..." I suggest rephrasing this so that it does not use the word "means". This will avoid potential confusion with the alternate definition of "means" that relates to averages. This will be helpful since the word "average" is used later in the sentence.

Thanks for the suggestion. This phrase is removed in the process of reconstructing the Methods section.

• Line 230: "positive S0 reflects ..." I suggest using a word other than "reflects" here to avoid potential confusion with the alternate meaning of "reflects" that is associated with radiation.

Suggestion taken; we use *"…in consistent with…"* instead.

• Line 371: Please report uncertainties. This is very important information because it facilitates comparison with other work, among other things.

We report these uncertainties in the supplementary material, as reporting these uncertainties for each of the subplots in Figs. 6-9 of the manuscript would make the main text really cumbersome.

**REVIEWER 3:**

Summary comment

This research builds upon a growing body of literature seeking to understand the causal relationship between cloud droplet number concentration, liquid water path, and cloud albedo. The analysis quantifies the impact of meteorology and frequency of occurrence in which stratocumulus clouds reside within brightening and entrainment regimes. This is useful to constrain model responses of stratocumulus clouds to changes in Nd. The paper is well written, and the concepts come across very clearly.

We thank the reviewer for the encouraging words.

However, my main concern rests on the methodology used to filter the data. First, the methodology needs some clarification to be reproducible and convincing.

We appreciate the reviewer's careful readthrough of the methodology section of this manuscript. In response to the reviewer's specific comments/suggestions, we have largely reconstructed the Methods section to enhance its clarity. **Detailed responses and revisions are listed following specific comments.**

Second, because the cloud albedo effect is sensitive to base state variables (LWP and Nd) it should be shown that filtering by cloud fraction, solar zenith angle, etc (necessary to remove untrustworthy satellite retrievals) does not adversely affect the population of the samples and bias the Nd-LWP-albedo relationship.

We have 3 main points to make regarding this comment:
i) the cloudy scenes (1° x 1°) that we analyzed in this study are not restricted to overcast conditions, while the CERES-MODIS footprints (20-km) are. Thus, our analyses do include partly cloudy scenes at the 1° x 1° scale **(see detailed responses below)**;
ii) because the $1° \times 1°$ cloud fractions of these cloudy scenes analyzed in this work are high, their contribution to the overall cloud radiative effect of the entire cloud population of this region is significant compared to the rest of the population (less cloudy). Thus, it is important and informative to quantify the response of these high-fc clouds to aerosol perturbations.
iii) more importantly, it is not the goal of this study to generalize our results to all marine stratocumulus clouds, especially those with thin optical depth, broken/open-cellular structures, or high sub-pixel inhomogeneity.
**Detailed responses and revisions are listed following specific comments.**

These changes may require major revision. Overall, I think this work makes a great contribution to the field. I have provided some comments and suggested changes below.

We thank the reviewer for these helpful suggestions/comments. The manuscript has been revised accordingly to clarify the concerns on the methodology and data filtering. Direct responses to comments and changes to the original manuscript are listed below.

Other comments

L4: Changes in "cloud fraction" are potentially as important, if not more so, than changes in LWP and Twomey on "modifying cloud radiative properties" to perturbations in Nd (e.g. see Goren and Rosenfeld, 2014, https://doi.org/10.1016/j.atmosres.2013.12.008).

We thank the reviewer for pointing this out. We reworded this as *"One of the important steps towards quantifying the role of aerosol in modifying cloud radiative properties is to quantify the susceptibility of cloud albedo and liquid water path (LWP) to perturbations in cloud droplet number concentration (Nd)."*

We also added a sentence in the Introduction section clarifying this point: *"In this work, we focus on the potential radiative impact of ``intrinsic'' cloud adjustments (due to changes in Nd and LWP). ``Extrinsic'' cloud adjustment (cloud fraction responses) is not addressed here."*

L6: It is difficult to relate this brightening potential to other papers and climate reports (e.g. IPCC) when the units are in W m−2 ln(Nd)−1. Can this be related to an effective radiative forcing by aerosol-cloud interactions in units of W m-2?

We agree with the reviewer that this radiative forcing per perturbation metric is hard to directly compare to existing aerosol-cloud interaction, effective radiative forcing estimates (e.g. IPCC), however, we elect to stick with this metric throughout this study for the following reasons:
a) there are existing studies that report this type of metric, e.g. Painemal (2018), Douglas and L'Ecuyer (2019), therefore, one can make a direct comparison, with appropriate caveats, as methodologies among these studies are not always identical,
b) this study focuses on high cloud fraction conditions over the northeast Pacific region, and we are not trying to generalize the results to global marine low clouds, therefore, even if we apply some form of Nd perturbation to yield a W m$^{-2}$ estimate, it would still be difficult to compare to global studies like the IPCC report, due to spatial scale differences,
c) such a global assessment extending the methodology of this study to global marine low clouds is currently in process, with plans to provide a global effective radiative forcing estimate, as a follow-up study.
d) the advantage of reporting a forcing sensitivity over reporting an actual forcing is that it represents the nature of the cloud itself, i.e. isolated from the underlying aerosol conditions, which highlights the potential of brightening/darkening in these clouds.

L8: Gryspeerdt et al. (2019), https://doi.org/10.5194/acp-19-5331-2019 discuss hypotheses involving similar regimes (entrainment-darkening and precipitation- brightening) as it relates to the Nd-LWP state-space. The word "identify" seems to suggest that this is a new discovery. I

would recommend that this specific wording be changed to something along the lines of "we discuss the results of two susceptibility regimes: the entrainment....".

We agree, and thanks for the suggestions. In fact, the abstract has been reconstructed to convey the main message and key findings of the study more clearly.

L14: Consider rephrasing the word "positively" – the word could be misconstrued as something that is "good" in this context.

The word "positively" has been avoided in the revised sentence, which now reads as *"… clouds that exhibit the strongest brightening potential occur …"*

Abstract: I think one of the novel results of this study is the analysis of cloud albedo in the Nd-LWP state space and that it exhibits a non-linear behavior as a function of Nd described in L201 – 205. I would recommend noting this behavior in the abstract and condensing the discussion on the intricate details with respect to meteorological drivers.

We agree with the reviewer on this point, and we did try to note this finding in the revised abstract. However, we ran into the issue of having too long an abstract, and noting this finding did not flow well with the main message on albedo susceptibility and meteorological influence Therefore, we choose to leave this out of the abstract.

Nevertheless, we include this finding as one of our key findings bullet points in Section 6.

L32: "quantify/constrain" which is it?

We rewrote this sentence to make it clearer, which now reads as *"…have been used to improve the quantification of the global aerosol radiative effect…"*

L34: While this is true, Boucher et al., 2013 is a bit out of date, Bellouin et al. (2020) would be better to cite here.

Reference is now updated to Bellouin et al. (2020) as suggested.

L37: "These processes occur at short timescales (order 5 – 10 min)" needs to be supported with a reference.

Reference added.

L72: I'm not sure the reference of Gryspeerdt et al. (2014) is appropriate here (saying that it is capable of "consecutive snapshots of an evolving cloud field"). They examined the time difference in MODIS between Aqua and Terra satellites (two points in time) of cloud systems and their change with AI. I would recommend citing a study that uses geostationary satellite observations with many more "snapshots" in time instead to make this point.

We thank the reviewer for pointing this out. Christensen et al. (2020) PNAS is now used for reference here.

Done.

Fixed.

L105: Why do you require such a strict criteria for $f_c > 0.99$? Grosvenor et al. (2018), shows that a threshold of 0.8 over 1-degree regions is sufficient to ensure the cloud field is homogenous. What effect does only including overcast clouds have on the analysis? I would think this could introduce a bias by removing partly cloudy cases where the clouds are likely to be more convective and possibly rainy. If $N_d$ is correlated to fc this may introduce an unwanted bias into the results.

First of all, the usage of strict criterion (0.99) at the footprint level (20-km) cloud fraction is to ensure more homogeneous cloud fields at which Nd is retrieved, even within stratocumulus regions, so that we are restricting to relatively high cloud fractions as measured over large areas, e.g. 1° x 1° (see Figure 4 above or Fig. S1 in the supplementary material), as suggested in Grosvenor et al. (2018);

We want to clarify that our strict fc > 0.99 criteria is applied at footprint (20-km) level, not at the 1-degree level (where we calculate albedo susceptibility), therefore, even though we only include overcast footprints, 1-degree cloud fraction can still be less than 1, meaning partly cloudy (1-degree) cases are included in our analyses, but restricted to high cloud fraction at the 1-degree scale. In fact, Figure 4 (included as Fig. S1 in the supplementary material) below shows the distribution of cloud fraction of 1-degree scenes used in our analyses, and more than 96% of the 1-degree scenes used in our study have a cloud fraction greater than 0.8 (**consistent with that used in Grosvenor & Wood (2014,2018)**), and only ~35% are overcast scenes at the 1 degree x 1 degree scale. For context and references, Painemal (2018) uses 0.95 at 20-km footprint and Gryspeerdt et al. (2019) use 0.9 at 5-km footprint.

I think the second point above clarifies that we did not remove all partly cloudy scenes in our analysis, but we did restrict to high cloud fraction scenes, thus, if Nd is correlated with fc as the reviewer pointed out, there is likely an "unwanted" bias in satellite-retrieved Nd representing the real climatology of Nd in nature. However, our current state-of-art satellite retrievals of Nd still need to rely on high cloud fraction scenes.

Regarding biases in the current results ($S_0$ characteristics) presenting the less cloudy scenes (more convective and possibly rainy). We now state that the validation of Nd-LWP-Ac relationships under heavily precipitating conditions is challenging with satellite snapshots, and we

acknowledge this as a caveat in the approach of this satellite observation-based (snapshot) study. Added discussion on this point now reads as *"One caveat associated with this approach is the difficulty in discerning the causal directions between Nd and LWP when the system is heavily precipitating and actively removing droplets from the system, as past states of the system cannot be obtained from polar-orbiting satellite snapshots. Because we focus on high cloud fraction scenes over a marine stratocumulus region, we expect heavily precipitating scenes to be rare in our analyses and assume the observed relationship between Nd and LWP under precipitating conditions reflects changes in the system if Nd were perturbed. We leave the validation of this assumption to a future evolution-oriented study that involves the temporal aspect of the cloud system."*

Last but not least, because the $1° \times 1°$ cloud fractions of these cloudy scenes analyzed in this work are high, their contribution to the overall cloud radiative effect of the entire cloud population of this region is significant, compared to the rest of the population (less cloudy). Thus, it is important and informative to quantify the response of these high-fc clouds to aerosol perturbations.

We added sentences at the end of the Datasets and the Method Sections to clarify this concern

*"As discussed further in Section 3, the fc > 0.99 condition at the CERES 20 km footprint allows for lower fc when these 20 km pixels are aggregated to $1° \times 1°$ scenes. For 1°x1° satellite sampled scenes, ~53% consist of single-layer liquid clouds only, and among these cloudy scenes, ~41% satisfy the Nd and $S_0$ calculation criteria (introduced in Section 3) and are subsequently used in this study."*

*"Note that requiring overcast conditions for Nd retrievals at footprint level does not restrict the $1° \times 1°$ cloudy scenes analyzed in this study to only overcast scenes, meaning partly cloudy scenes are included in our analyses. In fact, only ~35% of our 1° cloudy scenes are overcast (see the distribution of $1° \times 1°$ cloud fraction in Fig. S1). Because the $1° \times 1°$ cloud fractions of these cloudy scenes analyzed in this work are high (comprising ~41% of all single-layer liquid cloud scenes over this region), their contribution to the overall cloud radiative effect of the entire cloud population of this region is significant, compared to the rest of the population (less cloudy). Thus, it is important and informative to quantify the response of these high-fc clouds to aerosol perturbations. That said, it is not the goal of this study to generalize the albedo susceptibility assessment presented here to all marine stratocumulus clouds, especially those with low optical depth, broken or open-cellular structure (high sub-pixel inhomogeneity), conditions under which space-borne Nd retrievals are highly uncertain (Grosvenor et al., 2018)."*

L108: No justification is provided for the 600 cm$^{-3}$ threshold. Why?

A PDF of retrieved Nd before screening show that retrieving a Nd greater than 600 cm$^{-3}$ over the Northeastern Pacific stratocumulus region is extremely unlikely, with less than 0.01% likelihood (see Figure 5 below). Therefore, we conclude that Nd greater than 600 cm$^{-3}$ represents highly unrealistic retrievals, and thereby discarded. This justification is added in the revised manuscript accordingly as *"… (outside the 99.9$^{th}$ percentile) are discarded …"*

Also see Bretherton et al. 2010 ACP ([https://acp.copernicus.org/articles/10/10639/2010/acp-10-10639-2010.pdf](https://acp.copernicus.org/articles/10/10639/2010/acp-10-10639-2010.pdf)) for typical Nd values of marine stratocumulus over the southeast Pacific measured during VOCALS.

[Figure]

*Figure 5. Probability distribution function of retrieved Nd at 20-km CERES footprint.*

L130: "Moreover, although" is redundant, I would suggest removing "although"

Suggestion taken, "although" is removed.

L130 – 135: Is a longwinded sentence, can this be split into two? I'm also not clear after reading it why joint histograms built upon a composite of satellite snapshots better determine the conditional probability distributions. Better than what?

The Methods section has been largely re-written to enhance the overall clarity of the sentences introducing our approach, rationale behind choices, and assumptions we made. This original discussion on joint histograms is removed as a result of the revision to avoid confusion.

L140: Justification is needed. It is stated that a correlation of greater than 0.2 is required to limit "highly questionable and thereby unreliable" cloudy scenes but I don't understand why this argument is only applicable to low correlations? How many cases are being removed to fit this requirement? What about negative correlations (more negative than -0.2)? More information on filtering is necessary here to understand which cloud conditions are removed from the analysis so that the uncertainties are better understood.

We thank the reviewer for pointing this out, and we realize our original sentence was not very clear in justifying the choice. First of all, the correlation threshold is applied to the **absolute value**

of r (correlation coefficient), so, negative correlations are treated the same as positive correlations.

Second, we require a minimum number of samples (5) and an absolute value of r greater than 0.2 in order to bring up the significant level of these regressions while maintaining a reasonable number of cloudy scenes used for the analyses, which is a trade-off between increasing significant level and throwing away cloudy scenes.

With a correlation coefficient screening criteria of 0.2 (absolute value), we removed ~19% of the scenes from that without a |r| threshold. Figure 1 above shows the result without the |r(Ac,Nd)|>0.2 filtering, and one can see that this does not change the qualitative aspect of our results, but does change the overall occurrence-weighted $F_0$ a little from 17.0 (no |r| threshold) to 20.8 (|r|>0.2) W m$^{-2}$ ln(Nd)$^{-1}$.

The revised sentence now reads as

*"$S_0$ values are only reported if the number of data points is greater than or equal to 5 and the absolute value of the correlation coefficient if greater than 0.2. This provides levels ranging from 25% (minimum required number of samples, 5) to 60% (maximum number of samples within a 1° grid) at which the correlations are statistically significantly according to a Student's t-test. Applying such a threshold on the absolute value of the correlation coefficient between Ac and Nd shrinks the sample size of $S_0$ by ~19% but does increase the statistical significance of the results by at least 25%. A sensitivity test using $S_0$ without the correlation coefficient threshold (not shown) indicates no qualitative impacts on the results but a subtle quantitative impact on the occurrence-weighted $F_0$ (introduced below; from 20.8 to 17.0 W m-2 ln(Nd)-1)."*

L150: remove the apostrophe for the word sensors'

We actually think the apostrophe is necessary here.

L169 – L181: This discussion on filtering is unclear to me. The phrase "overcast footprints are weighted heavily over partially cloudy footprints" implies that the analysis uses clouds which have a cloud fraction of 0 < fc <= 1 but the next sentence says that Nd is only calculated when the footprint is overcast fc > 0.99. What I am unclear about is whether the data are grouped or ungrouped in this study. Do the population of clouds in calculations involving cloud albedo (equation 1) differ from those which use calculations that involve Nd (equation 2)? It may be that I am missing a subtle point here but some clarification would be helpful regardless.

We realize that this paragraph was indeed confusing and we reconstructed it to make it more concise and clearer. Basically, there is no grouping of data as far as 1-degree averaging is concerned, and for cloud properties, including cloud albedo, we are only using overcast footprints (20-km) for both $S_0$ calculations and 1-deg averaging, because Nd is only retrieved at overcast footprints.

This now reads as *"In order to understand and quantify how cloud albedo susceptibilities vary with changing cloud states (e.g. LWP, Nd), meteorological conditions, and aerosol loadings, we*

*aggregate cloud properties, including cloud albedo, and ERA5 meteorological variables (during the Aqua overpass over a 2-hour period) to the same 1° × 1° grid on which $S_0$ is calculated. The aggregation method follows a straightforward arithmetic mean of all the pixel-level data points within the grid (0.25° for ERA5 and 20-km for MODIS-CERES), except for cloud properties where we only select overcast footprints for averaging, because Nd is only retrieved at overcast footprints."*

Title: I would recommend adding the word "overcast" next to stratocumulus since these are the clouds which are being filtered in this study (as per my point above, unless I am wrongly misinterpreting the filtering of the data in this study).

As we clarified in one of the pervious comments from the reviewer, our methodology does not select only overcast cloudy scene (1-degree), in fact, we apply the overcast criteria at footprint level (20-km), and most of the cloudy scenes analyzed in our analyses have a cloud fraction greater than 0.8 (see Fig. 4 of this document). Therefore, we choose to stick with our original title.

L188: How is SZA "adjusted" to become 0 degrees. It is explained in the following sentence, but it is still unclear to me how you remove the seasonally varying SZA? Is the bias minimization function based on a theoretical or empirical calculation? Please explain.

The sentences on the SZA adjustment are revised to clarify the explanation, which now reads as *"Moreover, the cloud albedos used in this particular analysis are adjusted to an overhead solar zenith angle (SZA = 0°), in order to obtain a consistent basis for Ac-LWP-Nd relationships. This is done using the two-stream approximation (Meador and Weaver, 1980), which relates cloud albedo to cloud optical depth and solar zenith angle. Therefore, for a given τ we can obtain a theoretical Ac-SZA relationship using the two-stream approximation. The scattering asymmetry parameter is approximated by a linear function of re following Slingo (1989). We then use the theoretical τ-dependent Ac-SZA relationships to adjust Ac from measured SZA to overhead SZA."*

L233: Are these single-layer liquid "overcast" clouds?
L242: Again, can you confirm that these are overcast single-layer clouds?

No, these are single-layer liquid clouds with high cloud fraction (at 1-degree x 1-degree grid). Overcast condition is only applied to footprints of CERES-MODIS observations, as we stated above. We also clarify this in the main text, which now reads as *"It occurs ~22% of the time out of all the high cloud fraction single-layer liquid cloud we analyzed over the NE Pacific, based on this 10-year satellite-derived climatology." "… with negative $S_0$ occurs ~36% of the time out of the cloudy scenes we analyzed"*

Figure 1: "for all non-precipitating clouds (red)" while I understand this text refers to the slope value, consider changing it to a different color because OLWP < 500 g/m2 is also displayed and red and is slightly confusing.

Thanks for the suggestion. We changed the color of the slope line and value to magenta.

Figure 2: Can you say a little more about how the "Size of the filled circles in each panel indicates the relative frequency of occurrence" represents the data in this analysis? There appears to be a cutoff in which the diameter of one of the circles cannot be smaller than a certain size. What is that threshold and can it be included in the caption? Also, can "Occurrence-weighted mean radiative susceptibility" be clarified further? Presumably this means that larger circles will have more weight than smaller circles to the total radiative effect.

Thanks for this suggestion, and we added reference circle sizes to indicate corresponding frequency of occurrences and modified the caption to include this information.

Yes, the interpretation of the term "occurrence-weighted" is correct, and the clarified caption now reads as: *"…Mean radiative susceptibility ($F_0$) weighted by the frequency of occurrence of each LWP-Nd bin is printed in red (named "occurrence-weighted $F_0$")…"*

---

## Referee Report (RR1)

**Second Review of "Albedo susceptibility of Northeastern Pacific stratocumulus: the role of covarying meteorological conditions" by J. Zhang et al.**

Atmospheric Chemistry and Physics

December 1, 2021

The authors have addressed most of my comments from the first submission. However, I believe that the uncertainty analysis is still not entirely correct. For this reason, I recommend *minor revision* for the current manuscript, and I recommend that the manuscript be accepted for publication when this issue is addressed. Please see my comments on the uncertainty analysis below.

**Comments on Uncertainty Analysis**

If I understand the methods section correctly, the authors calculate spatial autocorrelation between CERES footprints *within* each 1°x1° lat-lon gridbox (line 196). If this is true, then the uncertainty quantification is not entirely correct because autocorrelation needs to be calculated for the variables that are used in the regressions (i.e. the 1°x1° gridbox-mean values, not the footprint values within gridboxes). The correct way to calculate spatial degrees of freedom is to first calculate gridbox-mean values of Ac. This will result in a three-dimensional array of Ac values with dimensions of lon, lat, and time. Then remove the climatological seasonal cycle from each lat-lon gridpoint and apply equation 5 of Bretherton et al. 1999 to the array to get the effective spatial degrees of freedom. I do not expect this to change the interpretation of the data that the authors have nicely presented, but I do think it is important that the uncertainty quantification is done properly so that the results can be compared to other studies.

---

## Author Response (AR2)

**Response to reviews of "Albedo susceptibility of Northeastern Pacific stratocumulus: the role of covarying meteorological conditions" by J. Zhang et al.**

The authors have addressed most of my comments from the first submission. However, I believe that the uncertainty analysis is still not entirely correct. For this reason, I recommend minor revision for the current manuscript, and I recommend that the manuscript be accepted for publication when this issue is addressed. Please see my comments on the uncertainty analysis below.

We thank the reviewer for the close read-through of the revised manuscript. Please see our response to the comments on the uncertainty analysis.

Comments on Uncertainty Analysis
If I understand the methods section correctly, the authors calculate spatial autocorrelation between CERES footprints within each 1°x1° lat-lon gridbox (line 196). If this is true, then the uncertainty quantification is not entirely correct because autocorrelation needs to be calculated for the variables that are used in the regressions (i.e. the 1°x1° gridbox-mean values, not the footprint values within gridboxes). The correct way to calculate spatial degrees of freedom is to first calculate gridbox-mean values of Ac. This will result in a three-dimensional array of Ac values with dimensions of lon, lat, and time. Then remove the climatological seasonal cycle from each lat-lon gridpoint and apply equation 5 of Bretherton et al. 1999 to the array to get the effective spatial degrees of freedom. I do not expect this to change the interpretation of the data that the authors have nicely presented, but I do think it is important that the uncertainty quantification is done properly so that the results can be compared to other studies.

We did calculate spatiotemporal autocorrelation between CERES footprints (20-km) within each 1°x1° lat-lon gridbox (line 196), and this is because the variables (Ac and Nd) used in the regressions are from CERES footprint-level (20-km), such that we use footprint-level Ac and Nd to obtain a regression slope for each 1°x1° lat-lon grid (we did not use 1°x1° lat-lon gridbox-mean values to perform the regression).

This was stated in lines between 161-165.

To clarify what spatial resolution of the variables actually go into the regression, we revised the sentence to read:
*"… we use slopes derived from least squares log-log regressions of 20-km footprint-level Nd and Ac, sampled by the MODIS and CERES sensors onboard the polar-orbiting Aqua satellite (1:30 local afternoon overpass)."*